

# Force Partitioning Analysis of Vortex-Induced Vibrations of Wind Turbine Tower Sections

Shyam VimalKumar[1,2], Delphine De Tavernier[1], Dominic von Terzi[1], Marco Belloli[2], and Axelle Viré[1]

[1]Faculty of Aerospace Engineering, Delft University of Technology, The Netherlands
[2]Department of Mechanical Engineering, Politecnico di Milano, Italy

**Correspondence:** Shyam VimalKumar (s.vimalkumar@tudelft.nl)

**Abstract.** Vortex-Induced Vibrations (VIV) of wind turbine towers during installation is an aero-structural problem of significant practical relevance. Vibrations may happen in the tower structure, especially when the rotor-nacelle assembly is not yet attached to the tower or if the rotor blades are not yet connected to the tower-nacelle assembly. The complexity of aeroelastic phenomena involved in VIV makes the modeling and analysis challenging. Therefore, the aim of the current research is to

investigate the fundamental mechanisms causing the onset and sustenance of vortex-induced vibrations. To gain more understanding of the nature of vibrations, a methodology is established that distinguishes different components of the forces at play. This approach allows identifying how various force components impact the oscillation of a rigid body. The method is executed using the OpenFOAM open-source software. Numerical simulations are conducted on a two-dimensional smooth cylinder at both subcritical and supercritical Reynolds numbers to establish a correlation between wind turbine tower vibrations and the

force mechanism. The analysis involves performing Unsteady Reynolds-Averaged Navier Stokes (URANS) simulations using the modified pimpleFoam solver with the k-$\omega$ SST turbulence model. Both fixed and free-vibrating cases are studied for smooth cylinders. For the high Reynolds number cases, a setup matching the tower top segment of the IEA 15MW reference wind turbine was chosen. Studying the flow around a cylinder at a subcritical Reynolds number reveals that the primary force involved is the vorticity force. The combined force due to viscosity, added mass, and vorticity contributes most to the overall

force. For a freely vibrating cylinder with a single degree of freedom in the cross-flow direction, the analysis indicates that the force component associated with the cylinder's motion is crucial and significantly affects the total force. Moreover, analysing the energy transfer between the fluid and the structure, a positive energy contribution by the vorticity-induced force is observed on or before the dominant Strouhal velocity. This confirms observations at low Reynolds numbers in the literature that the vortex shedding predominantly contributes to the initiation of oscillations during VIV. The kinematic force contributes to the

energy transfer of the system, but the mean energy transfer per cycle is negligible.

## 1 Introduction

Wind turbines continue to grow in size for energy production at lower cost of energy. One of the challenges that follow this trend is the scaling of wind turbines, as a taller wind turbines have longer and more flexible blades and towers (Hoen et al., 2023; Veers et al., 2019). Many physical phenomena must be taken into consideration during the design process for such large





turbines. Vortex-induced vibration (VIV) is one such aerodynamic phenomenon that must be understood in detail during the design of wind turbines. Both wind turbine blades and towers can suffer from VIV, for which many studies have been carried out in the past (Livanos, 2018; Derksen, 2019; Horcas et al., 2020; Riva et al., 2022). However, the complex aerodynamic interactions that give rise to VIV are not fully understood, especially for the complex flow conditions experienced by wind turbines. The towers are prone to oscillations or vibrations due to vortex shedding, particularly when the rotor-nacelle assembly

or the rotor blades are not yet installed. The load cases in which the wind turbine towers are susceptible to VIV are when the towers are: (i) standing on the quayside, (ii) transported in a vessel, and (iii) standing on the foundation (e.g. monopile or jacket for offshore turbines) either without the rotor-nacelle assembly or before the installation of the blades on the tower-nacelle assembly. Understanding VIV and its impact on wind turbine towers is crucial for the design and operation of these structures. Therefore, further studies are needed to analyse VIV and its mitigation for wind energy systems.

Extensive research has been conducted in the past to investigate VIV of circular cylinders, particularly for sub-critical and critical Reynolds numbers (Williamson and Roshko, 1988; Bishop and Hassan, 1964; Brika and Laneville, 1993; Williamson and Govardhan, 2004a). These studies investigated the vortex patterns and associated forces at various Reynolds numbers and the vibration pattern with respect to amplitude and direction. However, VIV is a complex phenomenon that depends on many other structural and fluid properties like mass, natural frequency, damping ratio, inflow velocity, turbulence intensity, viscosity,

etc. Whilst most of these studies have been carried out for low Reynolds number regimes, significantly fewer studies have been done for super- and trans-critical Reynolds numbers. This is mainly due to the difficulties of performing experiments at these high Reynolds numbers and the high computational expense of numerical simulations. Experiments carried out by Belloli et al. (2012, 2015) analyse the lift force behaviour and the wakes formed behind the cylinder. The cylinder is covered with a rough mesh to increase roughness and effective Reynolds number to perform these experimental analyses. Upon further investigation,

the motion experienced at a high mass ratio ($m^*$) and low structural damping shows that the vibration amplitudes are much higher than that according to the Griffin plot (Griffin, 1980).

In addition to the experimental studies, several numerical studies were carried out at trans-critical Reynolds numbers. These studies were performed using Large Eddy simulations (LES), Unsteady Reynolds Averaged Navier-Stokes (URANS) and hybrid RANS/LES approaches (Catalano et al., 2003; Squires et al., 2008; Travin et al., 2000). These studies analysed the

force and pressure around the cylinder at Reynolds numbers ranging from one to eight million. Some observations indicated significant disparities between the simulations conducted using LES (with wall-functions) and those performed using the URANS methodology. With the developments in hybrid RANS/LES, see (Fröhlich and von Terzi, 2008) for a review, the simulations have become simpler to execute as the resolution in the boundary layer on the cylinder could be significanlty reduced over wall-resolved LES. Detached Eddy Simulations (DES) and its variants, see (Spalart, 2009) for a review, is one

such method that showed great promise for cylinder flows. This is shown in work by Squires et al. (2008), where DES and Delayed DES (DDES) for flow over cylinders at transcritical Reynolds numbers are performed and compared with experimental data. Nevertheless, the more recent study by Viré et al. (2020) gives confidence in performing URANS simulations for trans-critical Reynolds numbers as envisioned here. Moreover, the studies were also carried out for oscillating cylinders under free and forced vibrations. Their results showed sufficient agreement for the purpose of the study between their URANS and work in



the literature, for both experiments and high-fidelity simulations. Most previous works analysed the numerical simulations for flow over cylinders at supercritical/ trans-critical Reynolds numbers for the total force and surface pressure over the cylinder. The vorticity and wake patterns are deduced qualitatively from their various mode shapes and shedding frequencies. But to better understand the fundamental mechanisms of initiation and sustenance of VIV, it is important to look into the details of the various force components and their relations with the cylinder motions.

By quantifying different force mechanisms acting on the cylinder, more insight into the behaviour of oscillations can be achieved. The work of Quartapelle and Napolitano (1983) proved to be a fundamental base for the force-partitioning method (FPM). A generic formula was derived for the force and moment of a rigid body immersed in incompressible flow. It was achieved by solving the Navier-Stokes equations where the momentum equation was projected on the gradient of a harmonic function, which satisfies appropriate boundary conditions. Chang (1992) further showed the force contribution due to potential flow, vorticity within the flow, and surface vorticity on the finite body, subsequently providing further understanding of different force components.

The influence of different forces on the oscillation of a rigid body provides more insight into the nature of the force mechanism. Morse and Williamson (2009) showed that the energy extracted in forced oscillations is closely related to amplitude response in free oscillations. This is found to be useful in complicated nonlinear responses and has been shown for oscillating cylinders by Williamson and Govardhan (2004b). This was combined along with the force-partitioning approach to analyse the force mechanisms in the sustenance of VIV by Menon and Mittal (2021). The latter found that the vortex shedding behind the cylinder contributes only during the initiation of VIV, whilst the vorticity in the shear layer leads to the sustenance of VIV. Hence, the vortex-induced force was found to be the most crucial for oscillations during VIV, compared to the vortices in the wake behind the cylinder. This finding is particularly interesting, especially for wind turbine towers, as the flow around wind turbines is characterised by a high Reynolds number ($Re > 3.5 \times 10^6$).

Even though VIV of circular cylinders has been extensively studied in the past, the study of VIV of cylinders under the influence of high Reynolds number flows, and large mass ratio is still limited. Moreover, as investigated by Viré et al. (2020), there is an interchange between self-exciting and self-limiting oscillation when the cylinder is allowed to vibrate freely. It is interesting to see how various force mechanisms contribute to such behaviour during VIV. Hence, the aim of the current work is twofold: (a) to implement the force-partitioning method in the open-source CFD software OpenFOAM (Bainbridge et al., 2012), and (b) to determine the nature of the force mechanism that leads to initiation and sustenance of VIV of wind turbine towers in the trans-critical Reynolds number regime. Additionally, it is intended to derive a correlation between the oscillations of the wind turbine tower and different force mechanisms using the above-mentioned approaches.

## 2 Methodology

### 2.1 Force partitioning method

The force-partitioning method aims to separate the forces experienced by a body in a fluid into various components in order to better understand the physical mechanisms between structural motions and the surrounding fluid flow. The method used in the



current study to analyse the various forces is derived from the previous works of Quartapelle, Mittal and Zhang (Quartapelle and Napolitano, 1983; Menon and Mittal, 2021; Zhang et al., 2015). This section explains the methodology used for the
partitioning of forces in detail.

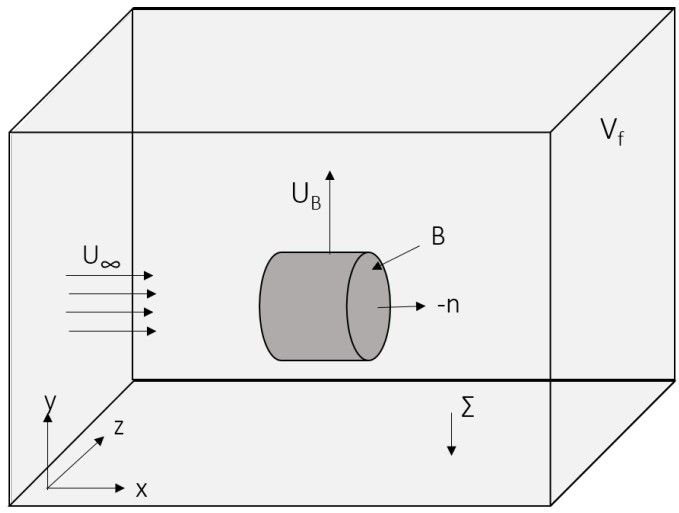

**Figure 1.** Schematic representation of a fluid domain containing a cylindrical rigid body.

Figure 1 shows the schematic representation of the domain. The surface of the body is represented as $B$, where the body is placed in the fluid domain $V_f$, which is bounded by an outer boundary $\sum$. The unit normal of every surface point of the object is represented as $n$. The method starts by introducing a harmonic function $\phi^{(i)}$, where $i \in \{1, 2, 3\}$ represents the direction in which the force is partitioned. The harmonic function corresponds to the potential associated with the flow around the body for
a shape and position for time $t = \tau$ as shown in Equation (1):

$$\nabla^2 \phi^{(i)} = 0, \quad \text{with} \quad \boldsymbol{n} . \boldsymbol{\nabla} \phi^{(i)} = \begin{cases} n_i, & \text{on } B(\tau) \\ 0, & \text{on } \Sigma \end{cases} \tag{1}$$

The derivation starts with the momentum conservation equation written in Lamb-Gromeka form, as shown in Equation (2):

$$\rho \frac{\partial \boldsymbol{u}}{\partial t} + \rho(\omega \times \boldsymbol{u}) + \frac{1}{2}\rho \nabla(\boldsymbol{u} \cdot \boldsymbol{u}) = -\nabla p - \frac{1}{Re}\mu \nabla \times \omega \tag{2}$$

where $t$ represents the time, $p$, $\rho$ and $\mu$ corresponds to pressure, density and dynamic viscosity, respectively. The partitioned
forces can be obtained by projecting this momentum equation on the gradient of the harmonic function, $\phi^{(i)}$, and integrating it over the fluid domain $V_f$, i.e.



$$\rho \int_{V_f} \left( \frac{\partial \boldsymbol{u}}{\partial t} + \boldsymbol{\omega} \times \boldsymbol{u} \right) \cdot \nabla \phi^{(i)} dV + \int_{V_f} \left( \frac{1}{2} \rho \nabla (\boldsymbol{u} \cdot \boldsymbol{u}) + \nabla p \right) \cdot \nabla \phi^{(i)} dV + \mu \int_{V_f} (\nabla \times \omega) \cdot \nabla \phi^{(i)} dV = 0 \tag{3}$$

Equation (3) upon further mathematical simplifications yields the final decomposition of the forces acting on the body $F_B$. This has been explained in detail in Menon et al. (Menon and Mittal, 2021). The final force is the summation of the

kinematic force ($F_K$), the vorticity-induced force ($F_\omega$), the viscous force ($F_\sigma$), the force that results purely from potential flow ($F_\phi$) and the force due to the effects of the outer boundary ($F_\epsilon$), as shown in Equation (4). The coefficient of each force component can be obtained by dividing the force component with the free-stream dynamic pressure, $q_\infty$, and the area $A$, namely $C_i = F_i / (\frac{1}{2} \rho U_\infty^2 A)$.

$$F_B = F_K + F_\omega + F_\sigma + F_\phi + F_\epsilon$$

$$F_B = \underbrace{-\rho \int_B \boldsymbol{n} \cdot \frac{d\boldsymbol{U_B}}{dt} \phi^{(i)} dS - \rho \int_B \frac{1}{2} U_B^2 \boldsymbol{n} \cdot \nabla \phi^{(i)} dS}_{\text{kinematic force}} + \underbrace{\rho \int_V \nabla \cdot (\boldsymbol{\omega} \times \boldsymbol{u}) \phi^{(i)} dV + \rho \int_V \nabla^2 \left( \frac{1}{2} u_\nu^2 + \boldsymbol{u_\phi} \cdot \boldsymbol{u_\nu} \right) \phi^{(i)} dV}_{\text{vortex-induced force}}$$

$$+ \underbrace{\mu \int_B (\boldsymbol{\omega} \times \boldsymbol{n}) \cdot \boldsymbol{\nabla}(\phi - \hat{e}_i) dS}_{\text{viscous force}} + \underbrace{\rho \int_V \nabla^2 \left( \frac{1}{2} u_\phi^2 \right) \phi^{(i)} dV}_{\text{potential flow force}} - \underbrace{\rho \int_\Sigma \frac{d\boldsymbol{U}}{dt} \cdot \boldsymbol{n} \phi^{(i)} dS + \mu \int_\Sigma (\boldsymbol{\omega} \times \boldsymbol{n}) \cdot \nabla \phi^{(i)} dS}_{\text{force due to flow and vortices at outer boundary}} \tag{4}$$

The velocity of the body at every point is represented as $U_B$, also shown in Figure 1. The velocity components $u_\phi$ and $u_\nu$ are obtained using the Helmholtz decomposition to expand the velocity vector into curl-free and divergence-free components as shown in Equation (5) (Batchelor, 2000):

$$\boldsymbol{u} = \boldsymbol{u_\phi} + \boldsymbol{u_\nu} = \nabla \Phi + \nabla \times \boldsymbol{A}, \tag{5}$$

where $\Phi$ and $\boldsymbol{A}$ are scalar and vector potentials, respectively.

## 2.2 OpenFOAM implementation

The numerical simulations for the present study are carried out using the open-source Computational Fluid Dynamics (CFD) software *OpenFOAM-v2012* (Bainbridge et al., 2012). OpenFOAM consists of many solvers and libraries, making it useful for various applications like fluid dynamics, structural dynamics, electromagnetics, etc. For the implementation of the Force-Partitioning Method (FPM), modifications made in the libraries of OpenFOAM are explained in the current section.

OpenFOAM consists of an incompressible solver named *pimpleFoam* which solves for pressure and velocity according to the PIMPLE algorithm. PIMPLE is a combination of the PISO (Pressure Implicit with Splitting of Operator) and SIMPLE (Semi-Implicit Method for Pressure-Linked Equations) algorithms. The incompressible solver is modified, named as *fppimpleFoam*, to calculate the velocity potential $\phi$ and the harmonic function $\phi^{(i)}$. However, to solve $\phi^{(i)}$, a boundary condition called





*fppotentialGradient* is made, which satisfies Equation (1). The variable $i$ (being equal to 1,2,3) partitions the force in the x,
y or z direction, respectively. The boundary condition is then given as input for the CFD simulation to consider the direction
in which the force is partitioned. The auxiliary function $\phi^{(i)}$ is introduced in the 0 folder, where the surface can be given the
boundary type of *fppotentialGradient* and the direction as (1 0 0) in order to partition the force in the x-direction. Finally, a
customized function object, *fpforceCoeff*, is derived from the *forces* class. This function object calculates the different force
coefficients for every time step according to Equation (4), which can be later accessed for post-processing. The equations
mentioned above work for an immersed boundary method. If a moving mesh is used to simulate the motion of the object, then
a correction term is to be introduced as shown in Equation (6):

$$F_{vorticity-correction} = -\rho\phi^{(i)} \int_B \boldsymbol{n} \cdot (\boldsymbol{\omega} \times \boldsymbol{U_B})dS \tag{6}$$

This term corrects for the influence of mesh motion in the Lamb vector as seen in Equation (2), thereby calculating the
rotational characteristics purely arising from the fluid flow. The complete flowchart of the implementation is shown in Figure
2 and the source code of the implementation can be found in the repository (VimalKumar, 2023).

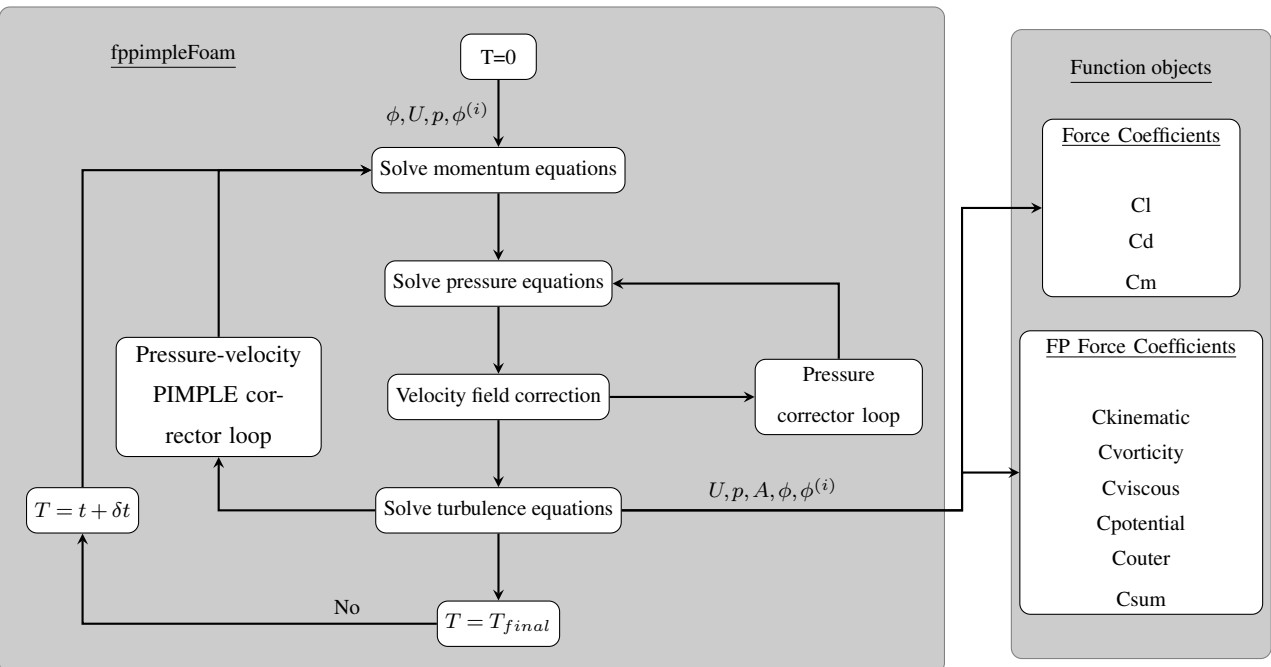

**Figure 2.** Flowchart for the implementation of the force-partitioning method in OpenFOAM.



## 3 Numerical setup

### 3.1 Governing equations and numerical methods

The numerical simulations in the current study are performed using *fppimpleFoam* together with the function object *fpforce-Coeff*. As it is a relatively simple geometry, a body-fitted Cartesian grid is made to carry out the simulations. The Unsteady
Reynolds-Averaged Navier-Stokes (URANS) equations are solved using the equations:

$$\nabla \cdot \overline{U} = 0,$$

$$\frac{\partial \overline{U}}{\partial t} + \overline{U} \cdot (\nabla \overline{U}) = -\frac{\nabla \overline{p}}{\rho} + \nabla \cdot (\nu \nabla \overline{U}) - \nabla \cdot (\overline{U'U'}) \tag{7}$$

The corresponding velocity and velocity fluctuations are represented by $U$ and $U' = U - \overline{U}$, where the overbar represents a suitable Reynolds average. For laminar simulations, $U' = 0$ and $U = \overline{U}$, and the equations revert to the traditional Navier-Stokes equations. The time derivatives in the above equation are solved using an implicit second-order backward scheme and
the spatial derivatives are computed using a second-order Gauss linear scheme. For the transcritical Reynolds numbers, the two-equation k-$\omega$ SST turbulence model (Menter, 1994) is used to compute the turbulent kinetic energy $k$, its specific dissipation rate $\omega$ and from these the eddy viscosity $\nu_T$. These are then used to close Equation 7 with

$$\overline{U'U'} = 2\nu_T(\nabla \overline{U} - 1/3\nabla \cdot \overline{U}) - 2/3kI, \tag{8}$$

where $I$ represents the identity matrix which is equivalent to Kronecker delta. A two-dimensional cylindrical section with a
spring-mass-damper system, as a way to represent wind turbine towers, is considered to analyse the fluid-structure interaction in this study. The spring and the damper are attached to the centre of mass of the cylinder. The spring has a spring constant of $k_s$ and the viscous damper has the damping coefficient $c$. The spring-mass-damper in this study gives the cylinder a single degree of freedom in the crossflow direction. The schematic representation of the system and the mesh are shown in Figure 3.

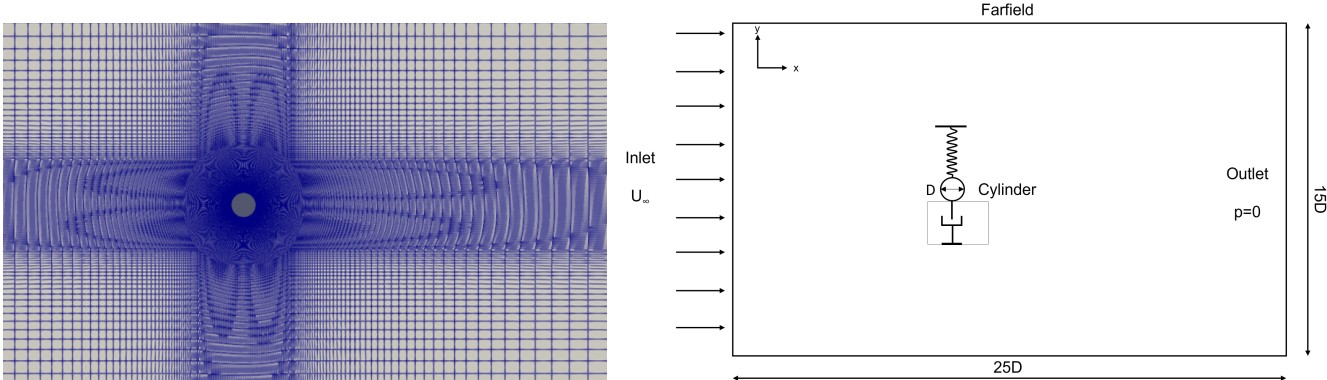

**Figure 3.** Schematic representation of the computational domain: Cartesian mesh (left), domain characteristics (right).



The governing equations for the motion of this rigid cylinder are represented as follows:

$$M\ddot{y} + c\dot{y} + k_s y = F_B \tag{9}$$

where $\ddot{y}$, $\dot{y}$ and $y$ represents the acceleration, velocity and displacement in y-direction respectively. The natural frequency of the system is determined by $\nu = \frac{1}{2\pi}\sqrt{\frac{k_s}{m}}$ for a damping ratio defined as $\zeta = \frac{c}{2\sqrt{k_s m}}$. The Fluid-Structure Interaction (FSI) simulations are carried out across various reduced velocities $U^*$ for a single mass ratio $m^*$. The reduced velocity is calculated

as $U^* = \frac{U_\infty}{\nu D}$ and mass ratio as $m^* = \frac{m}{\rho \pi D^2 L/4}$.

The structural properties of wind turbine towers must be calculated accurately to perform FSI simulations. With a given variation of mass density and stiffness, the natural frequency of the tower can be calculated. This is calculated from Rayleigh's method using the principle of equating maximum potential energy and maximum kinetic energy in the vibration mode (Clough

and Penzien, 1993). The wind turbine is considered as a stepped tower with $n$ segments to calculate the natural frequency. The displacement of the tower at any height $z$ and time $t$ can be written as

$$\nu(z,t) = \Psi(z) \cdot y_0 sin(\omega t) \tag{10}$$

where $\Psi(z)$ is the mode shape. The mode shape at any time represents the ratio of the displacement at the point $z$ to the reference displacement $y(t)$. As mentioned above, the maximum potential energy is equated to the maximum kinetic energy to

obtain the natural frequency. The potential energy of the system is given by

$$V = \frac{1}{2}\int_0^L EI(z)\left(\frac{\partial^2 \nu}{\partial x^2}\right)^2 dz \tag{11}$$

where $EI(z)$ is the flexural stiffness of the tower. The kinetic energy of the non-uniformly distributed mass is

$$T = \frac{1}{2}\int_0^L m(z)\dot{\nu}^2 dz \tag{12}$$

On substituting the displacement from Equation (10), the maximum potential and kinetic energy can be written as

$$V_{max} = \frac{1}{2}y_0^2 \int_0^L EI(z)\left(\psi''(z)\right)^2 dz$$

$$T_{max} = \frac{1}{2}y_0^2\omega^2 \int_0^L m(z)\left(\psi(z)\right)^2 dz \tag{13}$$

The natural frequency of the system can be obtained by equating $V_{max}$ and $T_{max}$ as

$$\omega^2 = \frac{\int_0^L EI(z)\left(\psi''(z)\right)^2 dz}{\int_0^L m(z)\left(\psi(z)\right)^2 dz} \tag{14}$$





The exact mode shape of a uniform beam of length $L$ undergoing vibration is taken as

$$\Psi(z) = \left(1 - cos\left(\frac{\pi z}{2L}\right)\right) \tag{15}$$

By considering the different mass density $m(z)$ over different turbine height $l(z)$ and substituting Equation (15) in Equation (14), we get

$$\omega^2 = \frac{\pi^4}{16L^4} \cdot E \cdot \frac{\sum\limits_{j=1}^{n} I_j l_j cos^2\left(\frac{\pi z_j}{2L}\right)}{\sum\limits_{j=1}^{n} m_j l_j \left(1 - cos\left(\frac{\pi z_j}{2L}\right)\right)^2} \tag{16}$$

Defining an equivalent moment of inertia and mass per unit length of the stepped beam,

$$I_{eq} = \frac{\sum\limits_{j=1}^{n} I_j l_j cos^2\left(\frac{\pi z_j}{2L}\right)}{L}$$

$$m_{eq} = \frac{\sum\limits_{j=1}^{n} m_j l_j \left(1 - cos\left(\frac{\pi z_j}{2L}\right)\right)^2}{L} \tag{17}$$

These equations are divided by the total length of the tower to ensure that both $I_{eq}$ and $m_{eq}$ have the right units of the moment of inertia and mass per unit length. Considering the tower top mass to be zero, Equation (16) then condenses to

$$\omega^2 = \frac{\pi^4}{16} \frac{EI_{eq}}{m_{eq}L^4} \tag{18}$$

### 3.2 Grid convergence study

The structured grid for the computational domain is generated using the *blockMesh* module of OpenFOAM and is shown in Figure 3 (left). Richardson's extrapolation method (Roache, 1994) is used for calculating a Grid Convergence Index (GCI) and quantifying the errors from the spatial discretisation of the mesh. This is a standardised method to analyse the numerical solution dependence on spatial discretisation. Three different grids are compared by having doubled the number of elements in the domain and carrying out h-refinement. The first cell height of the mesh, $y_{wall}$, is $1 \times 10^{-5}$ m, satisfying the near-wall dimensionless distance ($y^+$) to be less than or equal to 1 for every mesh. The domain consists of an O-grid of 5D diameter with the same degree of refinement throughout the meshes with a different number of elements. The ratio of the number of elements in a dimension between two meshes (coarse-medium and medium-fine) is adjusted to 1.26 in order to achieve an approximate total number of points around 2 between them. The details of the meshes are given in Table 1.

The pressure coefficient around the cylinder section is considered for the GCI study for a Reynolds number of 3.6 Million, as the paper mainly focuses on the VIV at trans-critical Reynolds number regimes. Figure 4b shows the coefficient of pressure along the cylinder surface of the medium grid with the error bars obtained using the GCI. The maximum error from the GCI study is found to be 2.45%, but the mean error is calculated to be 0.68%. The figure also shows the variation of pressure along the cylinder surface for various meshes in comparison to the study from Viré et al. (2020). It can be seen that there is



| Mesh | $y^+$ | Total No. of elements |
|--------|-------|-----------------------|
| Coarse | 1 | 96636 |
| Medium | 1 | 192880 |
| Fine | 1 | 385076 |

**Table 1.** Parameters of three different grids used in the grid convergence study.

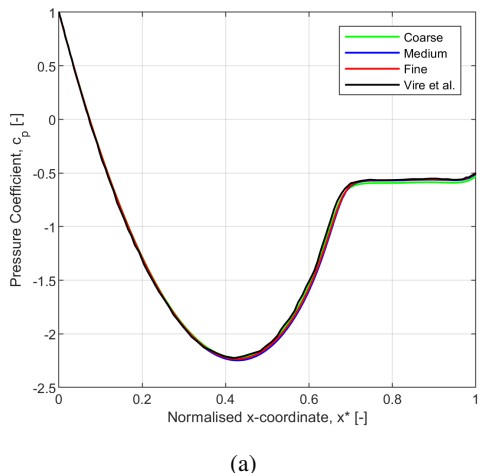

(a)

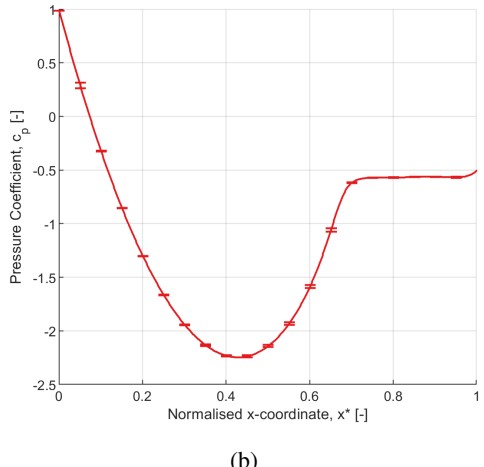

(b)

**Figure 4.** (a) Pressure coefficient along the surface of the cylinder section for coarse, medium and fine meshes in comparison to Viré et al. (Viré et al., 2020). (b) Pressure coefficient along the surface of the cylinder section with error bars using the GCI on the medium grid.

hardly any difference in the pressure along different meshes until the flow separates. However, the coarse mesh underpredicts the pressure coefficient after the separation point in comparison to other meshes. Furthermore, the medium and fine mesh predict the separation point as reported in the literature. Hence, the meshes are discretised with acceptable error, and further simulations are carried out with the medium mesh, as this mesh balances the complexity of the mesh motion and computational costs.

## 4 Results

Simulations are conducted for both laminar and turbulent Reynolds number regimes. Initially, simulations in the laminar regimes are performed to validate and verify the code implemented in OpenFOAM. Subsequently, simulations are carried out to replicate realistic wind conditions encountered by wind turbines. The focus of these simulations is primarily on free vibrations, mirroring real-world scenarios.



## 4.1 Flow in the laminar regime

A cylinder with a diameter of 2 m is considered for the simulations with sub-critical Reynolds numbers. The simulations are
performed for Reynolds numbers ranging from 80 to 200. The cylinder is considered to be rigid and steady, and hence, no
structural properties are defined. The force coefficients are then validated with results from the literature as shown in Figure 5,
where numerical simulations are performed using second order finite difference or volume methods (Park et al., 1998; Placzek
et al., 2009) and a spectral element method (Posdziech and Grundmann, 2007) . The lift and drag coefficients calculated
from the sum of the various force coefficients are marked as *Force-Partitioning Method*, and those obtained directly from the
OpenFOAM simulation are marked accordingly. It can be seen that the sum of the force coefficients matches well with both
the data from the literature and the present OpenFOAM simulation results.

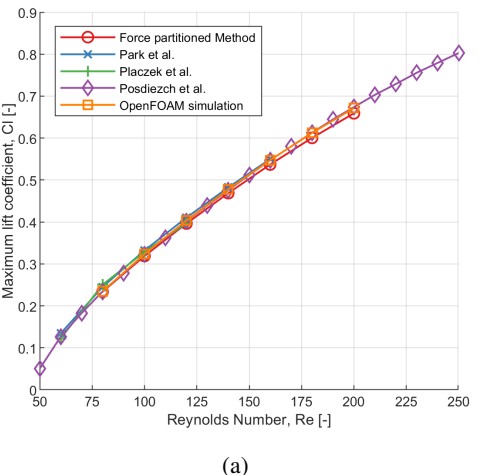
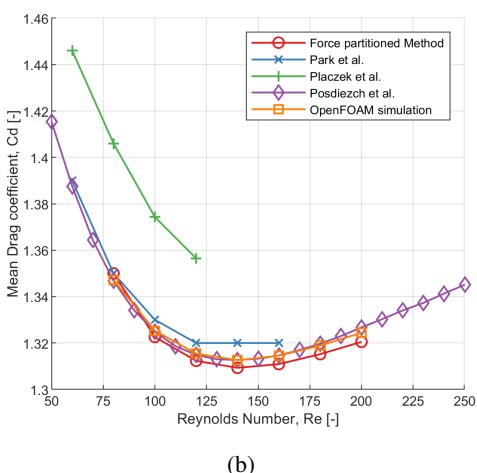

(a)  (b)

**Figure 5.** Variation of (a) maximum lift coefficient and (b) mean drag coefficient for a fixed cylinder at different sub-critical Reynolds
numbers compared against literature: citepPark1998, Placzek et al. (2009) and Posdziech and Grundmann (2007).

The flow around the cylinder subjected to motion in a crossflow direction is also validated and is shown in Figure 6. For a
mass ratio of 2 and a damping ratio of 0.007, the cylinder is subjected to a motion with various reduced velocities between 2
and 15 at a constant Reynolds number of 150. It can be seen that there is a jump in the amplitude at a reduced velocity of four,
where the frequency of vortex shedding matches the natural frequency of the cylinder. The maximum amplitude attenuates for
higher velocity ratios and approaching an almost constant amplitude from a velocity ratio of ten onwards. This matches well
with results in the literature, see (Carmo et al., 2011), for example.

The implementation of the force-partitioning method is verified against the calculation of the total force coefficient calculated
by OpenFOAM. The solvers in OpenFOAM are well validated and benchmarked, which is one of the main motivations for
verifying the code implementation with the OpenFOAM solver itself. Figure 7 shows the simulation results along with the
partitioned forces. The forces are partitioned in the y-direction,corresponding to the direction of the lift force on the cylinder.
As shown in the figure, the lift force matches well with the sum of the force coefficients. The slight difference between the

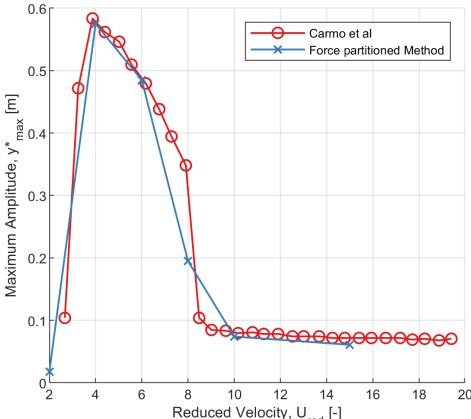

**Figure 6.** Comparison of the variation of non-dimensional maximum amplitude of a 2-D cylinder for various reduced velocities with results from Carmo et al. (2011); $Re = 150$, $m^* = 2$ and $c = 0.007$.

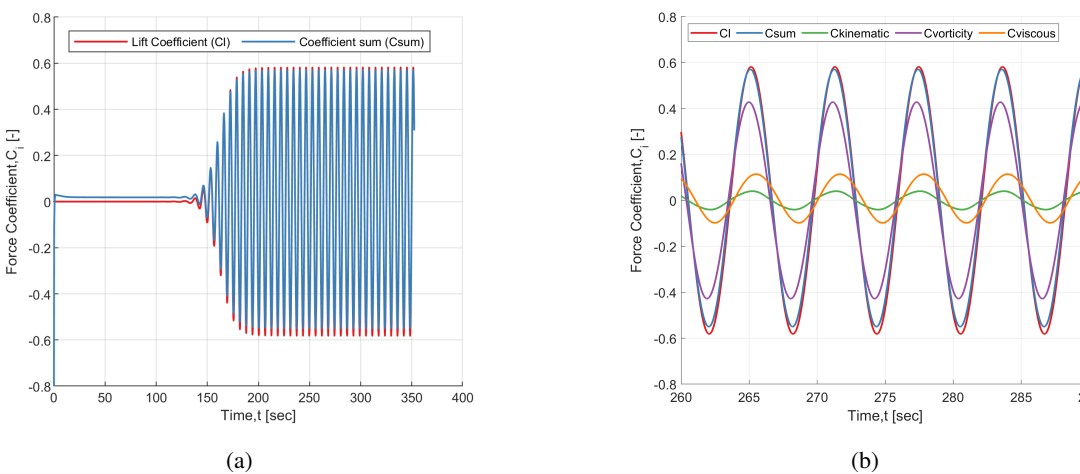

**Figure 7.** (a) Comparison of lift force coefficient against the sum of force coefficients from the force-partitioning method. (b) A zoomed-in snippet is shown, which breaks down the total force coefficient into partitioned force coefficients.

total lift coefficient and the sum of force coefficient is due to the discretisation errors in computing the forces with different methods (Viré and Knaepen, 2009).

When the total force is separated based on different coefficients, it becomes evident that the predominant factor influencing the force is the vorticity-induced force. The force due to the vortices formed behind the cylinder is in phase with the total force, and it drives to the total contribution. The other major contributors are the forces due to the motion of the cylinder and the viscous force. Depending on the oscillation frequency, the force could be in-phase or out-of-phase from the motion of the cylinder, which will be explained in more detail in Section 4.2.1. The results obtained above are in accordance with the results

observed by Menon and Mittal (2021) for sub-critical Reynolds numbers.



## 4.2 Flow in the turbulent regime

The laminar flow regime is far from a realistic scenario for wind turbine towers. Therefore, this section analyses the motion due to VIV of a section representative for the tower top of the IEA15MW reference wind turbine (Gaertner et al., 2020). Following the procedure explained in Section 3.1, the natural frequency of the tower is calculated to be 0.48. The parameters used for the simulation are mentioned in Table 2. The *fppimpleFoam* solver is employed to simulate and analyse the flow behind the 2D cylinder, which gives insights into the various forces acting on the cylinder for the transcritical Reynolds number regime. The cylinder has one degree of freedom in the y-direction. For the mass-damper system assumed for the motion of the cylinder, the parameters are tuned to obtain the required structural properties compiled in Table 2.

| Variables | Parameters |
|---|---|
| Diameter, $D$ [m] | 2 |
| Height, $z$ [m] | 1 |
| Reduced velocity, $U_{red}$ | 2.24 - 12.82 |
| Mass ratio, $m^*$ | 252.304 |
| Damping ratio, $c^*$ | 0.00318 |
| Reynolds number, $Re$ | $3.01 \times 10^6$ - $1.72 \times 10^7$ |
| Natural frequency, $f_{nat}$ [Hz] | 0.48 |

**Table 2.** Simulation parameters used for the free-vibration simulations of the IEA 15 MW reference wind turbine tower section.

In order to carry out the simulations for the turbulent Reynolds number regime, the $k-\omega$ SST turbulence model was chosen, as it is reported in the literature to be a good choice for flows with flow separation. The model also calculates the turbulent kinetic energy $k$ and the turbulent dissipation rate $\omega$ at every time step. The initial value of the turbulent kinetic energy is calculated as

$$k = \frac{3}{2}(TI \cdot U_\infty)^2,$$ (19)

where the turbulence intensity, $TI$, is taken as 0.03 and the inlet velocity is calculated depending on the reduced velocity used in the simulation. The turbulent dissipation rate is calculated as

$$\omega = \frac{\rho k}{\mu}\left(\frac{\mu_t}{\mu}\right)^{-1},$$ (20)

where $\frac{\mu_t}{\mu}$ is taken as 10 and $\mu$ is the dynamic viscosity of the fluid. The mesh parameters are explained in Section 3.2. The turbulence parameters at the cylinder boundary are taken as $k = 10^{-10} m^2/s^2$ and $\omega$ is calculated as

$$\omega = \frac{6\mu}{\beta_1 y_{wall}^2},$$ (21)

where $\beta_1 = 0.075$ (Menter, 1992).





### 4.2.1 Free vibration

As mentioned in Section 3.1, the simulations at turbulent Reynolds numbers are carried out with a single degree of freedom with translational motion in the y-direction. The cylinder has structural properties as mentioned in Table 2, and the simulations are performed for 12 different reduced velocities. The Reynolds number varies from three million to 17.2 million.

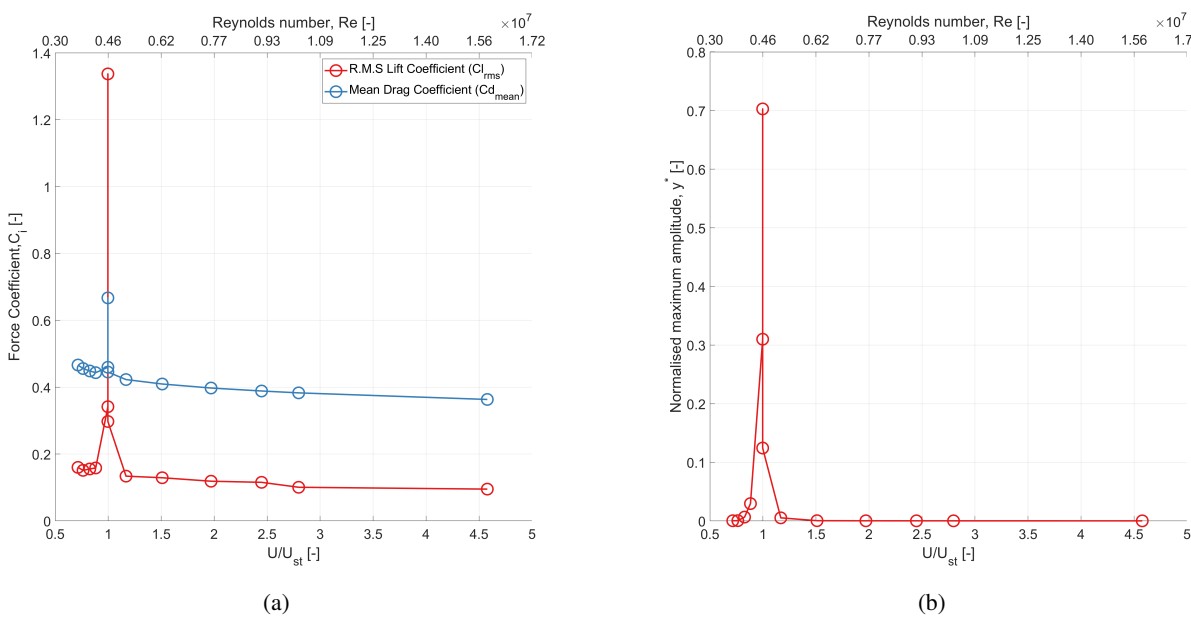

**Figure 8.** Variation of (a) root-mean square of lift coefficient and mean drag coefficient, and (b) normalised maximum amplitude for various velocity ratios and corresponding Reynolds numbers.

Figure 8 shows the variation of the force coefficients (a) and the maximum amplitude (b) for various velocity ratios. The Strouhal velocity is represented as $U_{st}$, which is defined as the wind velocity for which the vortex shedding frequency equals the natural frequency of the cylinder. The lift coefficient experiences a sharp rise when the velocity ratio $U/U_{st}$ reaches a value of unity. This also results in a peak in the normalised maximum amplitude of the cylinder, where it almost reaches 0.7. The normalised maximum amplitude becomes significantly less when $U/U_{st}$ is 1.2 or higher.







**Figure 9.** (a,c,e) Comparison of lift force coefficient with the partitioning coefficient at $U_{red} = 2.24$ ($U/U_{st} = 0.71$), $U_{red} = 2.88$ ($U/U_{st} = 0.997$), and $U_{red} = 3.20$ ($U/U_{st} = 1.05$), respectively. (b,d,f) Zoomed-in snippets of the time series of a,c,e, respectively.





Figure 9a and 9b show the behaviour of various partitioned forces at a reduced velocity 2.24. It is seen from Figure 9a that the main contributor to the lift force is the force due to vorticity. However, for the laminar (low Reynolds number) flow case, other significant contributors to the total force were present, i.e. the force due to added mass and the viscosity of the fluid. The zoomed-in figure from Figure 9b shows, that, for the turbulent (high Reynolds number) case without cylinder motion, the influence of the vorticity-induced force on the total force is the primary, or rather only significant, contributor to the total force.

As the free-stream velocity reaches the Strouhal velocity, the behaviour changes drastically, as seen from Figures 9c and 9e. As the lift force develops a beating pattern, the vorticity-induced force follows a similar behaviour. But the kinematic force is the dominating force overall. The magnitude of the lift force reduces mainly due to the out-of-phase behaviour between the kinematic force and the vorticity-induced force. This is a similar behaviour for both values of the velocity ratio: $U/U_{st} = 0.997$ and $U/U_{st} = 1.05$. The kinematic force is equally dominating as the oscillation sustains, but any reduction in the total lift force

is due to the out-of-phase trend of the two major force contributors, as can be seen in Figures 9d and 9f.

Figure 9 illustrates the changes in partitioned forces as the velocity ratio approaches and surpasses 1. The force coefficient amplitudes exhibit heightened values as the ratio nears unity, primarily attributed to the vorticity-induced force. However, reductions in amplitude are influenced by the kinematic force. This pattern reduces as the ratio exceeds 1.2.

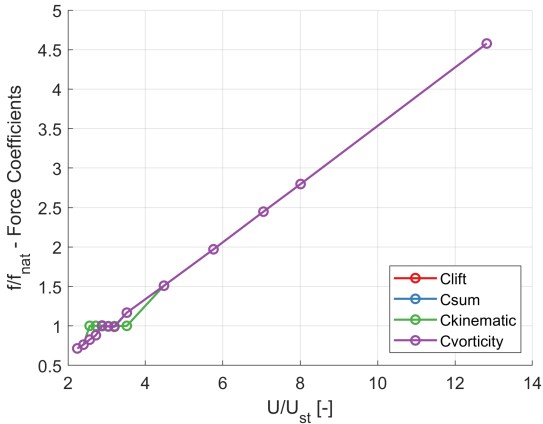

**Figure 10.** Frequency ratio of the forces as a function of normalised velocity $U/U_{st}$.

Figure 10 shows the frequency ratio of forces observed at various normalised velocities. It is observed that the kinematic

force fluctuates at the natural frequency of the structure, even before the structure starts to oscillate at its natural frequency. This provides the reason for the erratic fluctuations in the lift force. This behavior continues even after the natural frequency is reached until almost a frequency ratio of 1.2. This suggests that it may be important to analyse the phase difference of the forces to the displacement of the structure. Understanding the nature of the phase differences along with the energy mapping can give more insights into the nature of the oscillations experienced by wind turbine tower sections, as will be shown in the

next section.





### 4.2.2 Phase difference and energy transfer

The phase difference gives a first look into how energy transfers between the fluid and the structure. From Figure 11, it can be seen that there is a considerable phase difference between the lift force and the displacement of the cylinder as the velocity ratio reaches unity. By breaking down the contribution from the different forces, it can be seen that the vorticity-induced force is the

driving factor during this scenario. However, the phase difference between the kinematic force and the cylinder displacement is slightly negative when the velocity ratio is approximately unity. This could be interpreted as energy being added to the system primarily from the force due to the vortex shedding. This shows the influence of vortices in driving the oscillation, especially around the natural frequency of the system. Meanwhile, the viscous force has a constant negative phase difference with a sink around the velocity ratio of unity. This shows the damping nature of the force throughout, and it is significantly influenced

when the system oscillates at its natural frequency.

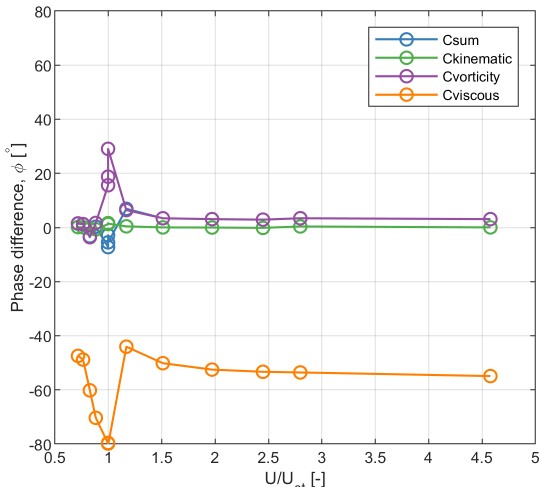

**Figure 11.** Phase difference between dominant forces and the displacement of the cylinder section for various velocity ratios.

A deeper analysis of the energy transfer between the fluid and the system provides more insights into the nature of each partitioned force. According to the literature (Morse and Williamson, 2009; Menon and Mittal, 2021; Kumar et al., 2016), the non-dimensional energy extracted by the cylinder can be written as

$$E^* = \int_t^{t+dt} C_i y^* dt, \tag{22}$$

where $C_i$ is the partitioned force, $y^*$ is the non-dimensional velocity of the cylinder and $dt$ is the time step. A positive energy transfer means that the system takes energy from the fluid, and vice-versa. Furthermore, this provides better insight into the major contributor to the motion of the cylinder, along with the initiation and sustenance of the oscillations. The energy transfer is analysed for the velocity ratios of interest ($U/U_{st} = 0.997, 1, 1.05$), which are close to the Strouhal velocity. Figures 12, 13





and 14 show the energy transfer per time step and the mean transfer during the oscillation cycles for the lift force, the kinematic
force, the vorticity-induced and the viscous force.

Figure 12 shows the energy transfer when the reduced velocity is 2.88 and the velocity ratio is 0.997. The major energy
contributor to the lift force is the energy from the vorticity-induced force. There is a similar pattern to the beating pattern
observed in the time series of the lift force. However, the mean non-dimensionalised energy from the vorticity-induced force is
positive, $E^* = 1.27 \times 10^{-4}$. Even though there is an overall contribution of energy from the kinematic force, the mean energy
transfer due to it is almost zero. As the fluid flow considered during the simulations is at high Reynolds number, the energy
from the viscous force is negligible as well.

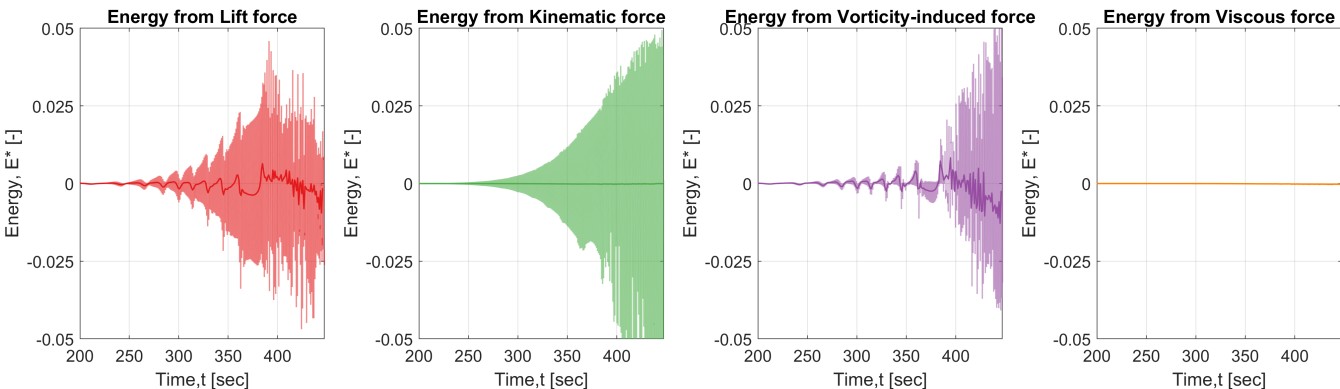

**Figure 12.** Non-dimensionalised energy transfer from lift force and partitioned forces between the fluid and the structure at a reduced velocity
of 2.88 ($U/U_{st} = 0.997$). The continuous dark line shows the moving average of the instantaneous normalised energy, while the shaded curve
shows the instantaneous value.

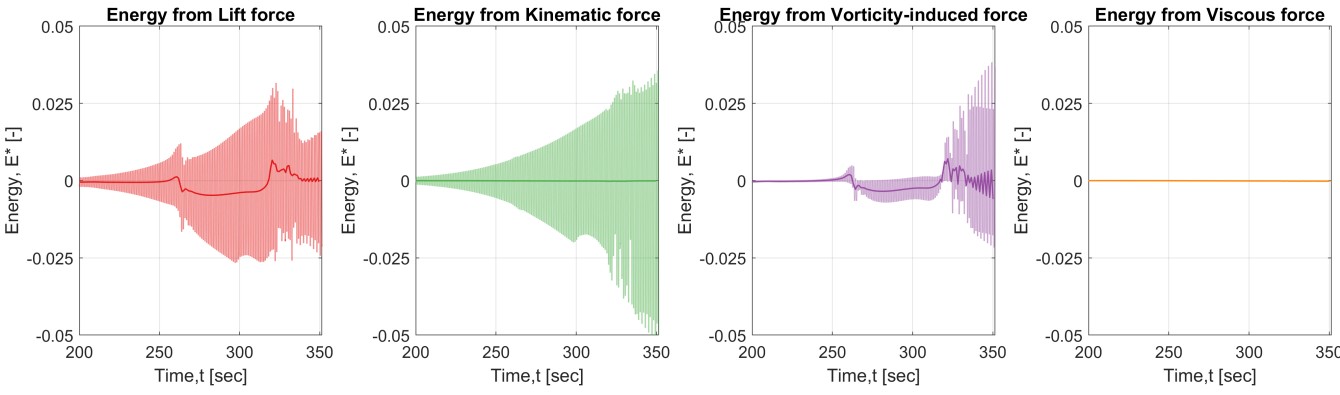

**Figure 13.** Non-dimensionalised energy transfer from lift force and partitioned forces between the fluid and the structure at a reduced velocity
of 3.04 ($U/U_{st} = 1$). The continuous dark line shows the moving average of the instantaneous normalised energy, while the shaded curve
shows the instantaneous value.



At a velocity ratio of unity, it is observed from Figure 13 that the transient nature of the flow is attained much quicker than for the other velocity ratios. The non-dimensionalised energy transfer happens from the structure to the fluid until a peak is observed, at $t \approx 320\ s$, before the completely transient nature of the system. Eventually, there is a net mean positive energy transfer from the fluid to the structure. This change in the nature of energy transfer may be due to the fact that the energy extracted from the shear layer increases as the cylinder oscillates at natural frequency, and that dissipated to the wakes is comparatively less, as explained in (Menon and Mittal, 2021).

For the velocity ratio $U/U_{st} = 1.05$, the net energy from the vorticity-induced force is largely negative, which implies that it starts to have a damping nature after the vibrations have been initiated. The rest of the oscillations are sustained by the energy from the added mass. As seen from previous figures, the mean energy contribution from the kinematic force is also negligible in this scenario. When the lift force is completely transient, it is observed that the vorticity-induced force increases as seen in previous conditions. Generally, the energy transition between the fluid and the structure shifts relatively swiftly to a fully transient state before reaching the Strouhal velocity.

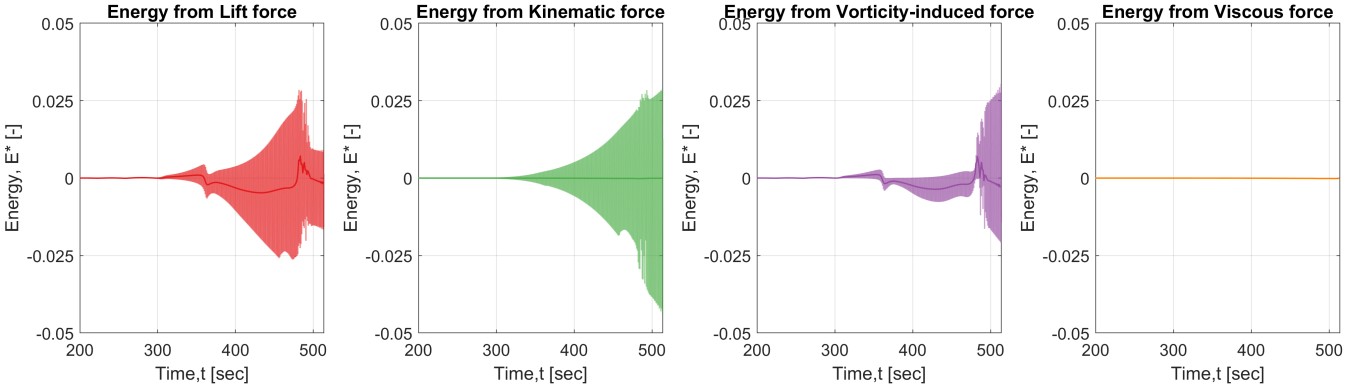

**Figure 14.** Non-dimensionalised energy transfer from lift force and partitioned forces between the fluid and the structure at a reduced velocity of 3.20 $(U/U_{st} = 1/05)$. The continuous dark line shows the moving average of the instantaneous normalised energy, while the shaded curve shows the instantaneous value.

## 5 Conclusions

In this current work, vortex-induced vibrations of a circular cylinder at a transcritical Reynolds number are analysed using the force-partitioning method. This method dissects the total force into various force components such as the force due to added mass, vorticity, the viscosity of the fluid, forces associated with the potential flow and the force due to the boundary of the domain. The methodology is implemented in the CFD software OpenFOAM, where the *pimpleFoam* is modified to *fppimpleFoam* and the force coefficients are calculated using *fpforceCoeff*. Unsteady RANS simulations were carried out to analyse the oscillation in crossflow direction using the turbulence model k-$\omega$ SST. The structural parameters were corresponding to the tower top section of the IEA 15MW reference wind turbine tower.





On analysing the decomposed forces at Reynolds numbers greater than $3 \times 10^6$, the major forces that contribute to the lift force are the vorticity-induced force and the kinematic force. When the cylinder oscillation is sufficiently away from the natural frequency ($U/U_{st} < 0.8$), the dominant force contributor is the vorticity-induced force. But as the frequency ratio increases,
the amplitude of the oscillation increases, which leads to increased contribution of the kinematic force. The patterns observed in the total lift force are mainly due to the phase difference between the kinematic force and the vorticity-induced force. The oscillations are initiated by the vortex formed behind the cylinder, consequently, the major force contributor in the initial time periods is the vorticity-induced force. The kinematic force oscillates at the natural frequency even before the system, which explains the transient fluctuations in the lift force.

The energy transfer between the system and the fluid gives a better idea of the nature of the force contribution. The mean energy contribution by the vorticity-induced force is positive on or before the Strouhal velocity. Therefore, the net energy transfer occurs from the fluid to the system. Even though there is a high energy contribution from the kinematic force, the mean energy transfer between the system and the fluid is negligible. As the oscillations are almost sinusoidal, the net energy per cycle becomes almost zero. Hence any transient behaviour or sudden change in lift force comes purely from the vorticity-
induced force. The energy contribution from the viscous force is negligible as the Reynolds numbers are high. This is in contradistinction to the low Reynolds number validation case investigated before.

Similar to the studies by Menon and Mittal (2021), even at trans-critical Reynolds numbers, the main contributor to the initiation and sustenance of the oscillations is the vorticity-induced force. In the present research, the analysis of energy derived from vortices spans the entire domain, while the impact of vortices in both near-field and far-field has not been considered,
which was explored in the previous study. As the shape and size of the cylinder matter predominantly in the oscillation, care must be taken in designing the wind turbine towers. The change in taper ratio and tower top diameter will also significantly influence VIV. The current work can be extended to analyse how the various force components vary for a three-dimensional tower. It is critical to understand how the vortices in the near-field and far-field affect the oscillations for a complete tower. Furthermore, the effect of the taper ratio should be investigated as a tapered tower potentially reduces the occurrence of VIV.
Finally, the influence of instabilities in inflow, like inflow turbulence and shear, on various force components and VIV is to be studied further.

*Code and data availability.* Force-partitioning method implemented in OpenFOAM is available in https://gitlab.tudelft.nl/svimalkumar/fpfoam. The datasets generated from the simulations, the post-processed results and the scripts are available in https://10.5281/zenodo.10529197.

*Author contributions.* SV carried out overall research under the guidance of DDT, DvT, MB and AV. The conceptualisation of the research
and the methodology was done by SV, DDT and AV. The implementation of the Force-Partitioning Method, the numerical simulations and the preliminary analysis were carried out by SV. The results were interpreted and visualised by SV, DDT, DvT, MB and AV. SV prepared the original draft, which was thoroughly reviewed and edited by DDT, DvT, MB and AV.



*Competing interests.* The authors declare that they do not have conflicts of interest.



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
