# Peer review of "Force Partitioning Analysis of Vortex-Induced Vibrations of Wind Turbine Tower Sections"

_Wind Energy Science, 2024_

## Author Comment (AC1)

**Force Partitioning Analysis of Vortex-Induced Vibrations of Wind Turbine Tower Sections**

WES-2024-10

May 2024

The authors would like to thank you for reviewing our draft paper, and for your valuable feedback and comments. In the following sections, we try to address the comments of each reviewer separately. The comments of the reviewer are marked in **bold** font, where the reply is written in *blue* and the changes in the manuscript as *red*.

**Reviewer 1**

**Major Comments**

**1. The manuscript is clearly structured and well-written. It contains almost all the information required to reproduce the work done (some suggestions provided below). However, the main criticism of this reviewer revolves around the novelty and impact of the performed work. It does not seem to be clear what the specific contribution of this work is. All the numerical methods, including the force partitioning approach, seem to have already been employed in previous work. Indeed, the FSI strategy employed seems to correspond to the one used in Viré et al. (2020), published in this journal, so that most of the manuscript seems to be somehow a duplicate. Along the same lines, the discussion derived from the application of the force partitioning did not seem to bring new information to the aeroelastic phenomena characterization, as it is uncertain to what extent the observations made are novel (knowing the amount of literature available through classical force decompositions). The authors are therefore encouraged to improve the writing of the manuscript, contextualizing how their findings are placed with respect to the state of the art, and how the new findings do impact what we already know about VIV.**

The authors would like to thank the reviewer for the comment. The novelty of the present study is two-fold, as mentioned below:

1. The force partitioning method allows the decomposition of total force into kinematic, vortex-induced and viscous forces, which helps to understand the driving force during the initiation and sustenance of VIV. The implementation of the methodology in OpenFOAM is the first of its kind, which is explained in the manuscript.

2. Contrary to the work of Viré et al. [4], the force partitioning method is used to analyse the force contribution during VIV at transcritical Reynolds numbers. Tower properties of IEA 15MW are used for the study, which was not the case in the previous literature.

According to the previous works of Belloli et al., [1], the surface roughness may be used to mimic the high Reynolds number phenomena at a considerably lower Reynolds number. This methodology can shed light on the contribution of surface roughness on VIV at lower Reynolds numbers in the future.

Most of the above-mentioned comment is explained in the last paragraph of the introduction section. However, to emphasize the novelty of this work further, additional text is added to the conclusion.

Line 80: While the previous study focuses on the flow around a cylinder at Reynolds number $\leq 250$, the importance of force partitioning method at transcritical Reynolds numbers is yet to be understood, as the flow around wind turbines is characterised by a high Reynolds number ($Re > 3.5 \times 10^6$).

Line 390: Even though force-partitioning analyses have been carried out in previous studies, investigating the phenomenon of VIV at such high Reynolds numbers with this method is the first of its kind.

Line 400: As mentioned in the previous section, surface roughness is imparted to mimic flows with high Reynolds numbers at a low Reynolds number range. Here, this methodology can shed a light on the contribution of surface roughness on VIV at lower Reynolds numbers.

**2. The rationale behind the 3D->2D casting should be further illustrated in the manuscript. The authors have selected a section corresponding to the tower top. Is it the most critical section with regards to a potential lock-in?**

As a wind turbine tower is a tapered structure, analysis of such a three-dimensional structure in two-dimensional is not straightforward. The purpose of simplifying the analysis is to study the force partitioning methodology on a simple geometry at high Reynolds numbers. Future studies will focus on numerical studies of complete three-dimensional towers. It is also to be noted that the simulations carried out in OpenFOAM are never completely two-dimensional, but the mesh is extruded with a single cell thickness in the third dimension.

The structural properties, such as mass and stiffness, vary along the height of a three-dimensional tower. Moreover, a wind turbine tower can be considered as a cantilever beam standing perpendicular to the ground. Hence, the bending characteristics of such a structure during VIV cannot be obtained in a two-dimensional study. To simplify the overall analysis, the section corresponding to the tower top is used for the present study, as the overall displacement is maximum at the tower top and not due to being a critical section in regards to a lock-in. The following changes are made to lines 265-268.

Line 265-268: The characteristics of the top section of the IEA 15MW reference wind turbine are chosen as the analysis of a three-dimensional structure in two-dimensional is not straightforward, and it simplifies the overall analysis. The tower top section is chosen as it experiences the maximum displacement in the scenario of VIV and is not due to being a critical section during a lock-in.

**3. This reviewer found problems seeing how the values of Table 2 (m*,c*) were computed,**

**and what their relation is to the mechanical properties and dimensions of the full tower.**

The mass ratio is calculated for the whole tower as shown below.

$$m^* = \frac{m_L}{(1/4) \times \pi \times \rho \times D^2}, \tag{1}$$

where $m_L$ is the mass per unit length, $\rho$ is the density of the air and $D$ is the average diameter considering the base and top. The mass ratio is calculated for the top section of the tower with a height of 12.386 m. The authors would like to thank the reviewer for identifying the typographical mistake in the value, which has been rectified to 87.6. With the consideration from another reviewer, $c^*$ is changed to $\zeta$ as explained in line 167. The damping for the tower is 2%, and the damping ratio is calculated with the following equation.

$$\zeta = \frac{\delta}{\sqrt{4\pi^2 + \delta^2}}, \tag{2}$$

where $\delta$ is logarithmic decrement.

**4. Regarding the FSI implementation, did the authors employ a weak or a strong coupling approach? How was the mesh deformed?**

In the present study, a strong coupling algorithm is used for both flows in the laminar and turbulent regime. The mesh deformation is carried out with Spherical Linear Interpolation Scheme (SLERP) interpolation which is already implemented in OpenFOAM.

Line 201: A strong coupling algorithm is utilised for the present study to give accurate results for modeling VIV. The structural parameters are computed for each iteration of the fluid solver, where the fluid solvers are sub-iterated for multiple iterations till the convergence criteria are met. The computed displacement of the cylinder moves the mesh according to Spherical Linear Interpolation Scheme (SLERP).

**5. Consider including a validation for the turbulent setup, or citing Viré et al. (2020) to that end.**

As explained in Section 3.2 of the manuscript, the grid-independent study is compared against the results observed by Viré et al. [4], where a similar simulation setup and study were performed. The present work now cites the previous work as extensive validation and a comparative study is made in the previous work. The following sentence is added to line 289, explaining the same.

Line 289: As explained in Section 3.2, the simulations performed at Reynolds number, $Re = 3.6 \times 10^6$, serve as a validation for the simulations at the turbulent regime. The force coefficient, pressure distribution, and separation point align with the results observed from the work of Viré et al. [4].

**6. Equation 10: The amplitude factor $y_0$ was never introduced. Even if not relevant for this work or for the derivation, the generalized form of the equation may also take an initial phase. I found the comment "The mode shape at any time represents the ratio of the displacement at the point z to the reference displacement y(t)." rather confusing. The mode**

shape is time-independent, and the reference displacement has not been introduced. The choice of axes notation is also confusing, as e.g. the axial direction of Figure 1 corresponds to "x".

The authors understand the confusion derived from the paragraph and the changes for lines 177-179 are made as shown below. Figure 1 is edited for better understandability.

Line 177: where $\Psi(z)$ is the shape function, which represents the ratio of the displacement at the point $z$ to the reference displacement $y(t)$. The reference displacement $y(t)$ is represented as a sinusoidal function with amplitude $y_0$.

**7. Statistics shown in Figure 8 may suffer from the short total times simulated. Could be interesting to include a comment in the manuscript, and to state if that could be a limitation for the analysis made.**

The authors would like to thank the reviewer for the comment, and the following text in red is added to line 315 of the manuscript. As an explanation, the simulations were stopped when the vorticity-induced force became fully transient in nature. The simulations conducted were free vibration, and the oscillations can reach even higher if there is insufficient negative aerodynamic damping. Due to a considerable amount of time, cost and the transient nature of the vorticity-induced force, the simulations were stopped at this particular behaviour.

Line 315: As seen in Figure 9, the simulations were stopped once the vorticity-induced force became completely transient in nature. The system becomes self-exciting and the fluid imparts energy to the structure if the structural damping coefficient is not enough to suppress the aerodynamic excitement. This can be analysed with a longer simulation time, which is out of scope for the current study due to increased time and cost.

**8. In Figure 12, for the moving average, was the considered window a multiple of the period of motion? In the opinion of this reviewer, that could facilitate future comparisons.**

The non-dimensional energy $(E^*)$ extracted by the cylinder is calculated according to equation 24 of the manuscript. The mean of $E^*$ is then calculated for every period of the energy cycle, which is a multiple of the period of force fluctuation.

**9. "As the oscillations are almost sinusoidal, the net energy per cycle becomes almost zero." This reviewer had problems understanding this statement. Even with pure sinusoids, a phase difference should lead to a non-zero energy transfer (when considering what happens at the frequency of motion). Note that, at the LCO of e.g. Figure 9.a, one should be "pumping" energy into the system to counteract the structural damping and maintain the same amplitude of vibration.**

The authors understand the confusion created by the sentence, and it is struck out to avoid further doubts. During each energy cycle, the mean non-dimensionalised energy transfer from kinematic force is negligible when compared to that of vorticity-induced force. Especially during the initiation of oscillations, much of the energy transfer happens from fluid to structure due to the vortices formed behind the cylinder. The

total force experienced by the cylinder is largely driven by this, which can also be seen in Figure 9c,e. Once the structural damping overcomes the forces derived due to vortices, the total force becomes out of phase with the displacement and beating phenomenon, as in Figure 9c, can be seen.

**10. While the distribution of the OpenFOAM implementation performed by the authors is highly appreciated, and it is well described in the manuscript, it could be interesting to improve the writing so that non-OpenFOAM practitioners do not get lost. For instance, one could include more "fundamental" descriptions of what the "0" folder contains, or how blockMesh generates the grids.**

In order to keep the manuscript related to the methodology and its implementation, the fundamental explanations of running OpenFOAM simulations are now updated in the readme of the GitHub repository. This is now explained in the manuscript in line 142.

Line 142: The source code of the implementation can be found in the repository, where the fundamental description of running the OpenFOAM simulation is explained in detail.

**Minor Comments**

**11. Could be interesting to comment on how the method should be extended to situations where the structure is characterized by more than one mode.**

It is expected to carry out such studies in the future with multiple structural modes and environmental conditions. As the simulations in OpenFOAM are not completely two-dimensional, the methodology or the implementation does not change. However, carrying out high-fidelity or mid-fidelity simulations can give an insight into which part of the structure gets excited and initiates the VIV, and vice-versa, and how VIV are damped. Breaking down the domain into different parts of the fluid domain also gives an insight into how the near wake and far wake regions affect the VIV, as can be seen from the studies of Menon et al. [3].

**12. Is "Cartesian grid" the best way to describe the mesh? I believe near the cylinder your grid lines become normal to the surface.**

It is now renamed to O-grid topology in line 147.

Line 147: As it is a relatively simple geometry, a body-fitted Cartesian grid mesh with O-grid topology is made to carry out the simulations.

**13. "Nevertheless, the more recent study by Viré et al. (2020) gives confidence in performing URANS simulations": while this is true, could passing to another turbulence model eventually have an impact on the conclusions of this work, for instance regarding the relative contribution of the forces?**

The present study uses k-$\omega$ SST as the turbulence model, proving to predict fluid separation and strong pressure gradients better. The present work matches well with the results from the previous experimental and numerical studies, which is the primary reason for choosing it as the turbulence model. Another turbulence model like k-$\epsilon$ seems to underpredict the skin friction with a similar boundary condition and is known to perform poorly on flows with large separation. The flow is found to have a delayed separation and lower base pressure, which eventually gave slightly higher drag as per the studies by Catalano et al. [2].

The physical parameters analysed from the numerical studies with different turbulence models can have different results for pressure coefficient, separation point, wakes, etc., which are inherent to the turbulence modelling itself. The relative contribution of forces could then vary depending on the above-mentioned physical parameters. However, the overall calculation of force, depending on the direction in which it is analysed, will be the same as global force parameters like lift and drag.

**14. Would the provided OpenFOAM modifications work for 3D cases in its present form?**

OpenFOAM is a code defined for three-dimensional space and the meshes are defined as such. To carry out a two-dimensional analysis, the mesh is made with a single-cell thickness, and suitable boundary conditions are provided at the patches that are normal to the direction of interest. Therefore, the code works perfectly for 3D cases in the present form.

**15. The link to Zenodo of the manuscript did not seem to be correct (but a working one was indicated by the authors in the submission).**

The URL is now changed to a working link.

`https://zenodo.org/doi/10.5281/zenodo.10529196.`

**16. Equation 9: the symbol used here for the natural frequency is later on used in the manuscript for displacements.**

The variable is changed from $\nu$ to $v$ for the equation 10, 11 and 12.

**17. Equation 11: for consistency with the rest of the derivation, maybe it is a good idea to use " for the second derivative.**

The equation is changed to maintain consistency as recommended.

**18. Equation 16: there is a jump in the derivation here, from continuous to discrete. Was it intended?**

As equation 14 explains, the natural frequency is obtained by equating potential energy and kinetic energy. On substituting the shape function to this equation, equation 16 can be obtained. One thing to consider here is that the differentiation term $\Psi^{''}$ is already carried out and the final form is written in equation 16.

**19. "frequency of the tower is calculated to be 0.48" -> 0.48 Hz**

The correction is now implemented.

**20. Figure 3: misses the coordinates for the center of the cylinder.**

Figure 3 is changed to include the upstream and downstream size of the domain, which is sufficient to determine the location of the cylinder in the domain.

**21. Some words on the computational cost of the simulations would be appreciated.**

The simulations are carried out with 20 cores, where one of the most expensive simulations required approximately 4950 CPU hours, for a velocity ratio of $U/U_{st} = 0.997$, and 2520 CPU hours, for $U/U_{st} = 0.71$, using Intel(R) Xeon(R) CPU E5-2670v2 processor. This is added to line 318.

**References**

[1] M. Belloli et al. "Vortex induced vibrations at high Reynolds numbers on circular cylinders". In: Ocean Engineering 94 (2015), pp. 140–154. ISSN: 0029-8018. DOI: `https://doi.org/10.1016/j.oceaneng.2014.11.017`.

[2] Pietro Catalano et al. "Numerical simulation of the flow around a circular cylinder at high Reynolds numbers". In: International Journal of Heat and Fluid Flow 24.4 (2003). Selected Papers from the Fifth International Conference on Engineering Turbulence Modelling and Measurements, pp. 463–469. ISSN: 0142-727X. DOI: `https://doi.org/10.1016/S0142-727X(03)00061-4`. URL: `https://www.sciencedirect.com/science/article/pii/S0142727X03000614`.

[3] Karthik Menon and Rajat Mittal. "On the initiation and sustenance of flow-induced vibration of cylinders: insights from force partitioning". In: Journal of Fluid Mechanics 907 (2021), A37. DOI: `10.1017/jfm.2020.854`.

[4] A. Viré et al. "Two-dimensional numerical simulations of vortex-induced vibrations for a cylinder in conditions representative of wind turbine towers". In: Wind Energy Science 5.2 (2020), pp. 793–806. DOI: `10.5194/wes-5-793-2020`. URL: `https://wes.copernicus.org/articles/5/793/2020/`.

**Reviewer 2**

**Specific Comments**

**1. The "subcritical" Reynolds number regime is generally understood to be different than the laminar Reynolds number regime (see for example the cited work by Belloli (2015)). In the present work, both terms are used interchangeably.**

In the present work, the simulations performed at subcritical Reynolds numbers are between 80 and 200. As the Reynolds numbers are low enough to form a laminar flow around the cylinder, the authors have mentioned it as a laminar Reynolds number regime. But, for a generalised term where the flow is no longer laminar (especially for Re > 250) but subcritical Reynolds number range, the term subcritical is used in the manuscript.

**2. Chapter 3.1: The reviewer is missing boundary conditions at the inlet for $k$ and $\omega$. There is information about $k$ and $\omega$ in lines 256ff but they are, at least for $k$, introduced as initial values.**

The authors would like to thank the reviewer for the comment. The following sentences are added to the manuscript at line 286.

Line 286: The boundary conditions for $k$ and $\omega$ are Dirichlet boundary conditions at the inlet with values equal to initial values as calculated above. Neumann boundary condition is imposed for both the variables at the outlet.

**3. Chapter 3.1: The reviewer is interested in the time steps used and the resultant Courant number. Has the used time step been verified?**

An adaptable time step is adopted for the simulations performed in the present study. A maximum Courant number of 0.7 is set to determine the time step for each time iteration, which is verified in the previous study by Derksen [3].

Line 234: The time step used for all the simulations maintains Courant–Friedrichs–Lewy (CFL) number equal to 0.7.

**4. Chapter 3.1: The reviewer is interested in the coupling approach of fluid and body.**

In the present study, a strong coupling algorithm is used for both flows in the laminar and turbulent regime. The following sentence is added to line 201, explaining the coupling algorithm.

Line 201: A strong coupling algorithm is utilised for the present study to give accurate results for modeling VIV. The structural parameters are computed for each iteration of the fluid solver, where the fluid solvers are sub-iterated for multiple iterations till the convergence criteria are met. The computed displacement of the cylinder moves the mesh according to Spherical Linear Interpolation Scheme (SLERP).

**5. Figure 3/numerical domain: Information about the size of the upstream and downstream**

**part of the domain is missing.**

Figure 3 is changed to include the upstream and downstream size of the domain.

**6. Figure 3/numerical domain: The domain in the across-wind direction seems rather small with 15 d and a blockage ratio that is above 7 %. The cited work by Viré (2020) uses 100 d in the across-wind and 100 d in the along-wind direction. Verification of the domain size can show the appropriateness of the chosen domain size.**

The blockage ratio from the current domain size is 6.67% (Frontal area of cylinder/Inlet area of the domain $= \frac{d}{15d}$). Even though the work by Viré et al. [6] has adopted 100d along-wind and across-wind direction, the domain size can be reduced for quicker calculations with more mesh points, without having reflections from the domain boundary. As mentioned in the literature study by Rodriguez et al. [5], the current blockage ratio falls within the values mentioned in the previous studies on VIV. Moreover, the separation point or transition point will not be affected as the wake width decreases as Reynolds numbers reach supercritical Reynolds number ranges [1].

The simulation is validated and verified for subcritical and transcritical Reynolds number regimes as explained in Section 4.1 and 3.2 respectively. The numerical simulations at Reynolds number $3.6 \times 10^6$ is validated against the work of Viré et al. [6], where satisfactory accuracy is obtained as mentioned in Section 3.2.

**7. Chapter 3.2: The grid convergence study is conducted for the surface pressures on a fixed cylinder in transcritical Reynolds number regime. The main aim of the paper, however, is to investigate the situations for a SDOF oscillator. In the reviewer's view, the grid convergence study should investigate a critical situation during free vibration. Additionally, the increase of number of cells by a factor of 2 seems rather small even for a 2D simulation.**

The authors believe that the Grid Convergence Index (GCI) conducted for surface pressures on a fixed cylinder is still scientifically relevant in the present study. While investigating the situations for a single degree of freedom oscillator, another complexity of mesh motion adds up in the GCI study, which cannot be separated from the discretisation error. Nevertheless, the comparison of mean surface pressure on a fixed cylinder not only confirms the global integrated variables like force coefficients but also the surface pressure at every point around the cylinder. Moreover, the validation done at subcritical and transcritical Reynolds numbers adds to the confidence of the result.

According to the procedure estimated by Celik et al. [2] and Roache [4], Richardson's extrapolation technique is used to estimate GCI. A grid refinement factor greater than 1.3 is suggested by the literature, which can be attained by doubling the number of mesh points. In the present study, the grid refinement factor is approximately 1.413.

**8. Table 1: The reviewer is interested which y+ value is shown. Is it the mean value?**

This is the value at which the first cell height of the mesh is calculated. The first cell height is kept constant for all the meshes, while the mesh growth ratio is changed.

**9. line 209f: How is the separation point calculated and where is it located?**

The point where the wall shear stress becomes zero is taken as the separation point on the cylinder. The separation points for three meshes are as follows: Coarse - 112.9 °, Medium - 112.2 ° and Fine - 111.8 °. The line 222 is modified as shown below.

Line 222: Furthermore, the medium and fine mesh predict the separation point to be 112.2° and 111.8° respectively, in comparison to 111° from the literature.

**10. Figure 5, 6, and 8: It is not clear how the statistical values are obtained. Is the whole time history used, or is an initial time of the simulation discarded?**

The mean drag coefficient and r.m.s of lift coefficient are calculated after the simulation achieved a steady state. However, the whole time series is considered for the calculation of the maximum lift coefficient and maximum non-dimensionalised amplitude of oscillation.

**11. Figure 5, 6, and 8: The reviewer is missing information about the verification of the convergence of the results. It is not clear how the chosen simulation time has been deemed appropriate to calculate the statistical values.**

The results obtained for flow in the laminar regime are concluded after developing steady-state oscillations. Hence the verification of convergence of results is considered reliable for all the simulations in this Reynolds number regime. But for simulations carried out at turbulent regimes, the simulations were stopped after vortex-induced force became fully transient. As the simulations performed were free vibration, the structural damping was insufficient to restrict the oscillation at this point in time, especially around the lock-in region. In all the other velocity ratios, especially when the oscillations were not as significant as lock-in, harmonic steady-state solutions were obtained. The following sentence is now added, line 315, explaining the same for flows in turbulent regimes.

Line 315: As seen in Figure 9, the simulations were stopped once the vorticity-induced force became completely transient in nature. The system becomes self-exciting and the fluid imparts energy to the structure if the structural damping coefficient is not enough to suppress the aerodynamic excitement. This can be analysed with a longer simulation time, which is out of scope for the current study due to increased time and cost.

**12. One of the main concerns of the reviewer are the already mentioned diverging forces in Figures 9 c) and e). To the best of the reviewer's knowledge, the lift force does not diverge during VIV. See for instance the wind tunnel results by the co-author Belloli (2012) and Belloli (2015). In the numerical simulations of the co-author Viré (2020) it is shown that the lift force decreases significantly after reaching a maximum in Figure 11 $\omega* = 0.95$. Is this change of lift coefficient in time a particular situation that only appears in URANS simulations or has it been already observed in wind tunnel or full-scale?**

As seen from Zasso et al. [7], there is an irregular development of lift coefficient at the beginning of the lock-in region, where a phase difference much greater than zero can be seen. After a certain time, the phase difference drops to almost zero, and the lift coefficient and the oscillation amplitude are perfectly

synchronized. However, this phenomenon is also seen in URANS simulations because there is a switch in phase difference from positive to negative. This is mainly due to the change in wake pattern, which eventually dampens the oscillation, which causes the force to shift from a self-exciting to a self-limiting effect [3]. This can also be seen in the present study with the phase difference between kinematic and vorticity-induced force, as explained in section 4.2.1 of the manuscript.

**13. The reviewer is interested to know if the oscillation amplitudes also diverge. VIV are self-limited and the oscillation amplitudes should reach a maximum even during lock-in, contrary to aeroelastic phenomena such as galloping or flutter.**

As seen from Figure 8, VIV is a self-limiting phenomenon where large oscillations occur near the lock-in region. Any higher or lower wind speeds result in little to no vibrations in the structure. In the current study, the maximum amplitude is reached at a velocity ratio of 1 at a Reynolds number of $4.6 \times 10^6$.

**14. Figures 9 c) and e): The reviewer is missing information about the criterion used to stop the simulations after 450 s and 525 s, respectively. What happens after the shown time histories?**

As explained in comment 11, the simulations were restricted as the vortex-induced force reached fully transient. The structural damping was insufficient to limit the oscillation at this point of time for simulations in the lock-in region. The simulations are also stopped as the computational time and cost became considerably high and the uncertainty of not knowing when an equilibrium is reached. The following sentence, as explained in comment 11, is added in line 315.

Line 315: As seen in Figure 9, the simulations were stopped once the vorticity-induced force became completely transient in nature. The system becomes self-exciting and the fluid imparts energy to the structure if the structural damping coefficient is not enough to suppress the aerodynamic excitement. This can be analysed with a longer simulation time, which is out of scope for the current study due to increased time and cost.

**Technical corrections**

**15. line 22f: "challenge" instead of challenges, "as taller" instead of "as a taller"**

The correction is made in the manuscript.

**16. Figure 4: The definition of the normalized x-coordinate x* is missing.**

The definition for normalised x-coordinate, as shown below, is explained in line 217.

Line 217: The normalised x-coordinate $x^*$ is the x-coordinate normalised to the chord of the cylinder, as $x/c$.

**17. The direction of lift and drag and the respective formulas for the coefficients could be defined for clarification.**

The lift and drag forces are defined in line 239 at the beginning of the results. The equations for lift and drag are shown in Equation 19.

Line 239: The lift and drag coefficients calculated from the OpenFOAM postprocessing function tool are explained in Equation 19, where $u$ and $v$ are the velocity components in the x-direction and y-direction, respectively. The lift and drag force are defined as $F_L$ and $F_D$, respectively, and the surface area of the cylinder as $S$.

**18. Figure 5 caption: remove "citep".**

The caption is edited for correction.

**19. line 236 "y-direction,corresponding" add space after comma.**

The correction is now implemented.

**20. Table 2: Damping ratio has been introduced as $\zeta$ on p.8 instead of $c^*$ here. Is there a difference between these two damping ratios?**

It was a typographical error from the authors and it is now changed to $\zeta$.

**21. line 264: $y_{wall}$ is not defined.**

The first cell height of the mesh is defined by the variable $y_{wall}$, which is defined in line 2.

**22. The dataset is not available at https://10.0.20.161/zenodo.10529197 .**

The URL is now changed to a working link.

```
https://zenodo.org/doi/10.5281/zenodo.10529196.
```

**References**

[1] H. Blackburn and George Karniadakis. "Two and Three-Dimensional Simulations of Vortex-Induced Vibration of a Circular Cylinder". In: *Proceedings of the Third (1993) International Offshore and Polar Engineering Conference* (Jan. 1993).

[2] Ismail B. Celik et al. "Procedure for Estimation and Reporting of Uncertainty Due to Discretization in CFD Applications". In: *Journal of Fluids Engineering* 130.7 (July 2008), p. 078001. ISSN: 0098-2202. DOI: 10.1115/1.2960953.

[3] Adriaan Derksen. "Numerical simulation of a forced and freely-vibrating cylinder at supercritical Reynolds numbers". Master's Thesis. TU Delft, 2019.

[4]  P. J. Roache. "Perspective: A Method for Uniform Reporting of Grid Refinement Studies". In: *Journal of Fluids Engineering* 116.3 (Sept. 1994), pp. 405–413. ISSN: 0098-2202. DOI: `10.1115/1.2910291`. URL: `https://doi.org/10.1115/1.2910291`.

[5]  I. Rodríguez et al. "On the flow past a circular cylinder from critical to super-critical Reynolds numbers: Wake topology and vortex shedding". In: *International Journal of Heat and Fluid Flow* 55 (2015). Special Issue devoted to the 10th Int. Symposium on Engineering Turbulence Modelling and Measurements (ETMM10) held in Marbella, Spain on September 17-19, 2014, pp. 91–103. ISSN: 0142-727X. DOI: `https://doi.org/10.1016/j.ijheatfluidflow.2015.05.009`.

[6]  A. Viré et al. "Two-dimensional numerical simulations of vortex-induced vibrations for a cylinder in conditions representative of wind turbine towers". In: *Wind Energy Science* 5.2 (2020), pp. 793–806. DOI: `10.5194/wes-5-793-2020`. URL: `https://wes.copernicus.org/articles/5/793/2020/`.

[7]  A. Zasso et al. "Pressure field analysis on oscillating circular cylinder". In: *Journal of Fluids and Structures* 24.5 (2008), pp. 628–650. ISSN: 0889-9746. DOI: `https://doi.org/10.1016/j.jfluidstructs.2007.11.007`. URL: `https://www.sciencedirect.com/science/article/pii/S0889974607001119`.

**Reviewer 3**

**Remarks**

**1. Within the text, the force related to shedding is called, alternatively, vorticity force and vortex-induced force. The authors should uniformise how they refer to it.**

The authors confirm the non-uniformity, and the word has been replaced with vortex-induced force throughout the manuscript.

**2. The added mass force, first term in the RHS of equation 4, is never properly defined. Moreover, while maybe obvious, the difference between the volume V and the surface B in integrals is not given in the text.**

The first term in equation 4 corresponds to the added mass effect. It is determined by the rate of change of velocity of the object. Hence, it is purely dependent on the velocity and shape of the object, as it is integrated over the surface $B$. The equation is briefly explained in lines 110-121, where the detailed explanation can be found in Menon et al. [2], as mentioned in line 111. The terms $V$ as volume and $B$ as surface are explained in lines 98 and 99. The equation is now changed to maintain consistency in the terms for volume from $V$ to $V_f$ as shown below.

$$F_B = F_K + F_\omega + F_\sigma + F_\phi + F_\epsilon$$

$$F_B = \underbrace{-\rho \int_B \vec{n} \cdot \frac{d\vec{U_B}}{dt}\phi^{(i)}dS - \rho \int_B \frac{1}{2}U_B^2\vec{n}\cdot\nabla\phi^{(i)}dS}_{\text{kinematic force}} + \underbrace{\rho \int_{V_f}\nabla\cdot(\vec{\omega}\times\vec{u})\phi^{(i)}dV + \rho\int_{V_f}\nabla^2\left(\frac{1}{2}u_\nu^2 + \vec{u_\phi}\cdot\vec{u_\nu}\right)\phi^{(i)}dV}_{\text{vortex-induced force}}$$

$$+ \underbrace{\mu\int_B(\vec{\omega}\times\vec{n})\cdot\vec{\nabla}(\phi-\hat{e}_i)dS}_{\text{viscous force}} + \underbrace{\rho\int_{V_f}\nabla^2\left(\frac{1}{2}u_\phi^2\right)\phi^{(i)}dV}_{\text{potential flow force}} \underbrace{-\rho\int_\Sigma\frac{d\vec{U}}{dt}\cdot\vec{n}\phi^{(i)}dS + \mu\int_\Sigma(\vec{\omega}\times\vec{n})\cdot\nabla\phi^{(i)}dS}_{\text{force due to flow and vortices at outer boundary}}$$

$$(1)$$

**3. Figures 1 and 3 are inconsistent. Moreover, if the cylinder is 2D, what is the height H in table 1? It would help to understand this parameter if it is added to figure 3.**

Figure 1 is changed to make it consistent with figure 3. It is to be noted that Figure 1 is more representative in nature and not the domain characteristics of the present study. The domain characteristics explained in Figure 3 are shown as two-dimensional, as the mesh extrudes with a single cell thickness, unlike a full-fledged three-dimensional mesh or numerical study. It is carried out in such a way as OpenFOAM is a code designed for three-dimensional space and cannot have two-dimensional mesh. The boundary conditions in the patches normal to the direction of interest are made to make the flow completely two-dimensional. This is explained in lines 162 and 273.

Line 162: As OpenFOAM is a code designed for three-dimensional space, the mesh is extruded in the z-direction with one cell thickness.

Line 273: The parameter *height* shows the single cell thickness in the z-direction as mentioned in Section 3.1.

**4. It is not clear which criteria were used to choose the set of parameters (c, EI, ks) from equations 9-18 (or the mass ratio in table 2).**

The structural parameters mentioned in equations 9-18 are calculated as follows:

Youngs modulus, E for IEA 15MW tower $= 2 \times 10^{11}$ N/m$^2$

Moment of inertia for each section is calculated as

$$I_j = \frac{\pi(D_{\text{out}}^4 - (D_{\text{out}} - D_{\text{in}})^4)}{64}, \tag{2}$$

where $I_j$ is the moment of inertia of a section, $D_{out}$ is the outer diameter of the cylinder, $D_{in}$ is the inner diameter of the cylinder. The equivalent moment of inertia is calculated as mentioned in equation 17 in the manuscript.

Spring constant, $k$ is calculated from the equation

$$k = m\omega^2, \tag{3}$$

where $m$ is the mass of the tower/cylinder.

The parameters of the tower were chosen from the IEA 15 MW report as per IEA Task 37 [1]. The mass of the cylinder with a height of 1 m is calculated from the mass ratio of the IEA 15 MW tower. Equation 2 mentioned above is added to line 193 in the manuscript and the equation 3 is mentioned in line 167.

Line 193: where the moment of inertia of a section of the tower is calculated as

$$I_j = \frac{\pi(D_{\text{out}}^4 - (D_{\text{out}} - D_{\text{in}})^4)}{64}, \tag{4}$$

and $D_{out}$ is the outer diameter of the cylinder, and $D_{in}$ is the inner diameter of the cylinder.

**5. Why no comparison with the literature has been made for the cylinder in the turbulent regime? This could be done, at least, for a non-oscillating system. This would be still relevant as the turbulent wake may be conditioned by the turbulence model, separation point,etc.**

A grid-independent study is carried out in the present study, where the force coefficient, pressure distribution and its difference are simulated for various meshes. As seen in Section 3.2 of the manuscript, the results are compared with the works of Viré et al. [3], where a similar simulation setup and study was performed. An extensive validation and comparative study was made by Viré et al., and the present work is referenced to their works, as the results align with the previous study. The following sentence is added to line 289, explaining the same.

Line 289: As explained in Section 3.2, the simulations performed at Reynolds number, $Re = 3.6 \times 10^6$, serve as a validation for the simulations at the turbulent regime. The force coefficient, pressure distribution, and

separation point align with the results observed from the work of Viré at al. [3].

**6. Overall, the labels in figures are too small and hard to read even in electronic form.**

The figures are made bigger for better legibility.

**References**

[1]  Evan Gaertner et al. *Definition of the IEA 15-Megawatt Offshore Reference Wind Turbine.* Tech. rep. International Energy Agency, 2020. URL: `https://www.nrel.gov/docs/fy20osti/75698.pdf`.

[2]  Karthik Menon and Rajat Mittal. "On the initiation and sustenance of flow-induced vibration of cylinders: insights from force partitioning". In: *Journal of Fluid Mechanics* 907 (2021), A37. DOI: `10.1017/jfm.2020.854`.

[3]  A. Viré et al. "Two-dimensional numerical simulations of vortex-induced vibrations for a cylinder in conditions representative of wind turbine towers". In: *Wind Energy Science* 5.2 (2020), pp. 793–806. DOI: `10.5194/wes-5-793-2020`. URL: `https://wes.copernicus.org/articles/5/793/2020/`.

---

## Referee Report (RR1)

**Overall comments**

The paper's objectives are well defined and the sections clearly outline the authors goal. However, the text is not easy to follow to due long sentences. Also, the work performed to implement the code in OpenFOAM is a major feat which needs to be praised and recognised.

My recommendation is to publish the paper once the following comments have been addressed.

**Technical comments**

- L31: the list of possible ViV is interesting, it will be better to illustrate the impact thanks to, ideally, a picture of it happening or at least a sketch
- Fig3, why the cylinder not centred in the domain ? Was there any sensitivity study performed regarding the domain size and /or cylinder position ?
- L170: precise the FSI is performed using modal approximation (similar to OpenFAST). At first, it could be understood that a FE approach is done.
- L174: illustrate the "stepped tower" used in your calculation
- Eq10 and Eq15 use capitalised psi ($\Psi$), while Eq11,12,13,14 use the non-capitalised version ($\psi$)
- Eq11 and Eq12: consistency in writing the derivative. Either $v''\, and\, v'$ or $\dot{v}\, and\, \ddot{v}$; here both are used
- L195 / Eq18, the phrasing "equivalent moment of inertia and mass" can be misleading as it looks like an average value, since it is the total sum divided by the length ?
  Moreover, saying "These equations are divided by the total length of the tower to ensure that both i_eq and m_eq have the right units of the moment of inertia and mass per unit length" feels like the i_eq and m_eq are build to satisfy your needs rather than the opposite. If it is not the case, could you please elaborate ?
- L218: the mean error is calculated with respect to the Richardson extrapolation or previous results from Virée et al. ?
- L223: the separation point is provided in degree, I suppose it is when the cross section of the cylinder is plotted on an r-theta coordinate system ? Here everything is provided in cartesian or in x/c coordinate. Could you translate your results from degree to x/c system, so that it fits the plot in Fig4
- L269: The methodology to derive the natural frequency is detailed in section 3.1 and the numerical result is 0.48Hz. Using the methodology described in "aerodynamics of wind turbine" from Martin Hansen. Do you end up to that same frequency ?
- Table 2: Understood that the code needs a 3D structure, however a single cell of 1m seems prone to introduce edge effect ? Have you checked that ?
  Why not introducing either a small cell size (e.g. y+ dimension) ? Or using the same refinement in the z-direction than for x and y but on a smaller length (to limit the computational cost)
- Section 4.2.1: It seems that GCS was performed only for Re = 3.6e6 ? I fear that for very high Re (e.g close to 18e6) the mesh may not be sufficient, have you performed analysis to verify that ? (even if it is not properly described in the paper)

- Fig9: very interesting plots! For consistency can you use the same "time zones" in your zoomed in snippets ?
- L306: I was expecting to see vortex shedding behind a cylinder when reviewing the paper. When discussing about the "beating pattern" it would be interesting to see it by plotting the vorticity is Q-criteria. With a cylinder having Von Karman streets should be easy to see and correlates to the shedding frequency.
- Fig11: it is a very important figure which introduce the core of the result. Can you add a zoomed in version next to it where you focus between 0 < U/U_st < 2 and disregard the viscous component
- L341, "vice versa" leads to confusion I don't think it is necessary here
- Fig12: I disagree with the interpretation of these plots. The energy transfer is analogous to the system's work and rather than the average the integral should be calculated. When applying the "work-energy principle" $W = \Delta E_k = \frac{1}{2}mv_2^2 - \frac{1}{2}mv_1^2$ where v_2 and v_1 are the cylinder speed after and before the work is done, you realise that W is non zero. The reason being the cylinder does not finishes at rest.
  Therefore, I would say, that the same conclusions drawn for Fig13 and Fig14 apply here. The instantaneous values seems the most relevant in the case of the building up to the ViV state. Looking at Fig12b and 12c, I would say that the kinematic force contributes more than the vortex induced force.
  It is even more obvious in Fig13 and Fig14, where the red shaded area is driven by the green shaded area
- The conclusion is well written and summarise nicely the work performed and outcomes. It is mentioned that tower designer should take care of cylinder size, taper, taper ratio. Will an second part of the current paper address in more details the relationships between those design parameters and the lock-in phenomena ?

---

## Author Response (AR2)

**Force Partitioning Analysis of Vortex-Induced Vibrations of Wind Turbine Tower Sections**

WES-2024-10

July 2024

The authors would like to thank you for reviewing our draft paper, and for your valuable feedback and comments. In the following sections, we try to address the reviewer's comments separately. The comments of the reviewer are marked in **bold** font, where the reply is written in *blue* and the changes in the manuscript as *red*. The line numbers are marked according to the updated manuscript.

**Technical Comments**

**1. L31: the list of possible ViV is interesting, it will be better to illustrate the impact thanks to, ideally, a picture of it happening or at least a sketch**

A figure is added in the introduction section showing the possible scenario when a wind turbine tower experiences VIV.

Line 33: Figure 1 shows a load case when a tower is standing on the foundation during installation at the site.

**2. Fig3, why the cylinder not centred in the domain ? Was there any sensitivity study performed regarding the domain size and /or cylinder position ?**

The cylinder is not placed in the centre but slightly toward the inlet, as is common in other literature [2, 5]. This further helps reduce the total number of cells required for the computation, thereby reducing costs.

A sensitivity study for the domain size was not performed for the current study but results are validated across a wide range of Reynolds numbers, further providing confidence on the domain size and cylinder placement.

**3. L170: precise the FSI is performed using modal approximation (similar to OpenFAST). At first, it could be understood that a FE approach is done.**

The sentence is modified as shown below:

Line 171: The structural properties of wind turbine towers must be calculated accurately to perform FSI simulations, which are carried out using modal approximation.

**4. L174: illustrate the "stepped tower" used in your calculation**

A figure is now added with edited text in line 174, as shown below.

Line 174: The wind turbine is considered as a stepped tower with $n$ segments to calculate the natural frequency, as shown in Figure 5.

**5. Eq10 and Eq15 use capitalised psi ($\Psi$), while Eq11,12,13,14 use the non-capitalised version ($\psi$)**

The equations are corrected and $\psi$ is used for consistency.

**6. Eq11 and Eq12: consistency in writing the derivative. Either $v''$ and $v'$ or $\dot{v}$ and $\ddot{v}$; here both are used**

The variable $v'$ represents the derivative of $v$ in space, whereas $\dot{v}$ represents the derivative of $v$ in time. Following sentence is added to line 185 for further clarity.

Line 185: , where $'$ represents space derivative and $\dot{}$ represents derivative in time.

**7. L195 / Eq18, the phrasing "equivalent moment of inertia and mass" can be misleading as it looks like an average value, since it is the total sum divided by the length ? Moreover, saying "These equations are divided by the total length of the tower to ensure that both $i_{eq}$ and $m_{eq}$ have the right units of the moment of inertia and mass per unit length" feels like the $i_{eq}$ and $m_{eq}$ are build to satisfy your needs rather than the opposite. If it is not the case, could you please elaborate ?**

Thank you so much for the comment. The term equivalent is used instead of average as $I_{eq}$ and $m_{eq}$ represent the moment of inertia and mass per unit length of a tower with varying dimensions along the length. As pointed out by the reviewer, line 199 may lead to confusion, and it is consequently removed.

Line 199:

**8. L218: the mean error is calculated with respect to the Richardson extrapolation or previous results from Viré et al. ?**

The mean error mentioned in line 220 is calculated with respect to the Richardson extrapolation. The error value is calculated between the medium and fine meshes. Furthermore, the meshes are compared to the literature, where an acceptable error in pressure curves and the separation point is found.

**9. L223: the separation point is provided in degree, I suppose it is when the cross section of the cylinder is plotted on an r-theta coordinate system ? Here everything is provided in**

cartesian or in x/c coordinate. Could you translate your results from degree to x/c system, so that it fits the plot in Fig4

The location at x/c coordinate is added to line 223, and the line is modified as shown below:

Line 223: Furthermore, the medium and fine mesh predict the separation point to be $112.2°$ ($x/c = 0.6889$) and $111.8°$ ($x/c = 0.6857$) respectively, in comparison to $111°$ ($x/c = 0.6792$) from the literature.

**10. L269: The methodology to derive the natural frequency is detailed in section 3.1 and the numerical result is 0.48Hz. Using the methodology described in "aerodynamics of wind turbine" from Martin Hansen. Do you end up to that same frequency ?**

The natural frequency of the tower is derived using the method to estimate the first flap-wise eigenmode, as described in the referenced literature. The frequency is calculated to be 0.56. The authors believe the reason for the difference in frequency could be twofold: (i) The natural frequency is calculated for the tower without including the waterline, but from where the tower actually starts. (ii) According to Structural Dynamics by Roy R. Craig [3], there can be a slight difference in the calculation of frequency between the Rayleigh method and other methods, depending on the Rayleigh Quotient calculated.

**11. Table 2: Understood that the code needs a 3D structure, however a single cell of 1m seems prone to introduce edge effect ? Have you checked that ? Why not introducing either a small cell size (e.g. y+ dimension) ? Or using the same refinement in the z-direction than for x and y but on a smaller length (to limit the computational cost)**

The single cell thickness is made to be in the similar order of the cell size near the inlet, outlet and far-field boundaries. The mesh is found to have sufficient quality when checked for aspect ratio, orthogonality and overall mesh parameters.

**12. Section 4.2.1: It seems that GCS was performed only for Re = 3.6e6 ? I fear that for very high Re (e.g close to 18e6) the mesh may not be sufficient, have you performed analysis to verify that ? (even if it is not properly described in the paper)**

The grid convergence study is indeed performed for $Re = 3.6 \times 10^6$, but the results are verified for $Re = 8 \times 10^6$ by comparison with the literature as shown in Table 1. The Reynolds numbers considered for this verification were up to $Re = 8 \times 10^6$. This seems reasonable as the wind turbine tower usually does not experience Re greater than $1 \times 10^7$ (corresponding to a wind speed of 25 m/s).

| Study | Re | Strouhal Number | Mean Drag coefficient |
|---|---|---|---|
| Derksen [1] | $8 \times 10^6$ | 0.34 | 0.36 |
| Squires et. al [4] | $8 \times 10^6$ | 0.37 | 0.37 |
| Present study | $8 \times 10^6$ | 0.34 | 0.397 |

Table 1: Comparison of results with the previous studies for $Re = 8 \times 10^6$.

**13. Fig9: very interesting plots! For consistency can you use the same "time zones" in your zoomed in snippets ?**

Thank you for your comment. The figures are updated with the same time limits on the x-axis.

**14. L306: I was expecting to see vortex shedding behind a cylinder when reviewing the paper. When discussing about the "beating pattern" it would be interesting to see it by plotting the vorticity is Q-criteria. With a cylinder having Von Karman streets should be easy to see and correlates to the shedding frequency.**

A new figure is added to the manuscript showing the vorticity behind the cylinder. The following lines are added to explain the figure.

Line 313: Figure 12 shows the vorticity formed behind the cylinder when the lift force develops a beating pattern. Figures 12a and 12b show the wake pattern at maximum positive displacements in a cycle, where Figure 12a depicts the time when the vortex-induced force is in phase with the kinematic force. However, Figure 12b illustrates the wake pattern when the two forces have a significant phase difference. It can be observed that vortices are shed from the bottom part of the cylinder when the forces are in phase. In contrast, vortices are shed from the top of the cylinder when the kinematic force and vortex-induced force are out of phase. The wake pattern development observed here is similar to that reported in previous studies by [6].

**15. Fig11: it is a very important figure which introduce the core of the result. Can you add a zoomed in version next to it where you focus between $0 < U/U_{st} < 2$ and disregard the viscous component**

A zoomed-in figure is added along with the existing figure.

**16. L341, "vice versa" leads to confusion I don't think it is necessary here**

The word is removed.

**17. Fig12: I disagree with the interpretation of these plots. The energy transfer is analogous to the system's work and rather than the average the integral should be calculated. When applying the "work-energy principle" $W = \Delta E_K = \frac{1}{2}mv_2^2 - \frac{1}{2}mv_1^2$ where $v_2$ and $v_1$ are the cylinder speed after and before the work is done, you realise that W is non zero. The reason being the cylinder does not finishes at rest. Therefore, I would say, that the same conclusions drawn for Fig13 and Fig14 apply here. The instantaneous values seems the most relevant in the case of the building up to the ViV state. Looking at Fig12b and 12c, I would say that the kinematic force contributes more than the vortex induced force. It is even more obvious in Fig13 and Fig14, where the red shaded area is driven by the green shaded area**

Authors agree to the reviewer's comment, and changes are made to the manuscript as shown below. The mean energy transfer gives an idea about the nature of the force in general. Moreover, the instantaneous energy transfer (or energy transfer for a cycle) gives a better understanding of the development of VIV. As per Figure 12 (Figure 15 in the updated manuscript), the energy contribution from the vorticity-induced force is significant during onset, which is now added as shown in the text below. However, in all the cases, the contribution from the kinematic force drives the oscillation to be sustained, as shown in Figures 13 (Figure 16 in the updated manuscript) and 14 (Figure 17 in the updated manuscript).

Line 355: The major energy contributor to the lift force during the onset of VIV is the energy from the vortex-induced force.

Line 358: As the oscillations become significantly larger, the energy from the added mass contributes to the overall energy in the lift force.

**18. The conclusion is well written and summarise nicely the work performed and outcomes. It is mentioned that tower designer should take care of cylinder size, taper, taper ratio. Will an second part of the current paper address in more details the relationships between those design parameters and the lock-in phenomena ?**

Thank you so much for your comment. The next steps in the research will be to carry out numerical simulations for a three-dimensional tower and to study the effect of surface roughness, as explained in the conclusions section of the manuscript.

**References**

[1] Adriaan Derksen. "Numerical simulation of a forced and freely-vibrating cylinder at supercritical Reynolds numbers". Master's Thesis. TU Delft, 2019.

[2] Muk Chen Ong et al. "Numerical simulation of flow around a smooth circular cylinder at very high Reynolds numbers". In: *Marine Structures* 22.2 (2009), pp. 142–153. ISSN: 0951-8339. DOI: `https://doi.org/10.1016/j.marstruc.2008.09.001`. URL: `https://www.sciencedirect.com/science/article/pii/S0951833908000403`.

[3] Jr. Roy R. Craig. *Structural Dynamics - An Introduction to Computer Methods*. John Wiley  Sons, 1981.

[4] Kyle D. Squires, Vivek Krishnan, and James R. Forsythe. "Prediction of the flow over a circular cylinder at high Reynolds number using detached-eddy simulation". In: *Journal of Wind Engineering and Industrial Aerodynamics* 96.10 (2008). 4th International Symposium on Computational Wind Engineering (CWE2006), pp. 1528–1536. ISSN: 0167-6105. DOI: `https://doi.org/10.1016/j.jweia.2008.02.053`. URL: `https://www.sciencedirect.com/science/article/pii/S0167610508000299`.

[5] R.M. Stringer, J. Zang, and A.J. Hillis. "Unsteady RANS computations of flow around a circular cylinder for a wide range of Reynolds numbers". In: *Ocean Engineering* 87 (2014), pp. 1–9. ISSN: 0029-8018. DOI: `https://doi.org/10.1016/j.oceaneng.2014.04.017`. URL: `https://www.sciencedirect.com/science/article/pii/S0029801814001565`.

[6] A. Viré et al. "Two-dimensional numerical simulations of vortex-induced vibrations for a cylinder in conditions representative of wind turbine towers". In: *Wind Energy Science* 5.2 (2020), pp. 793–806. DOI: `10.5194/wes-5-793-2020`. URL: `https://wes.copernicus.org/articles/5/793/2020/`.